# Bacterial ubiquitin ligase engineered for small molecule and protein target identification

James S Ye [ID] [1,11], Abir Majumdar [ID] [1,2,11], Brenden C Park[1], Miles H Black [ID] [1], Ting-Sung Hsieh [ID] [1,2], Adam Osinski [ID] [1,2], Kelly A Servage[1,2], Kartik Kulkarni[3], Jacinth Naidoo [ID] [4], Neal M Alto[5], Margaret M Stratton[6], Dominique Alfandari [ID] [7], Joseph M Ready[4], Krzysztof Pawłowski [ID] [1,2], Diana R Tomchick [ID] [4,8] & Vincent S Tagliabracci [ID] [1,2,9,10] ✉

## Abstract

The *Legionella* SidE effectors ubiquitinate host proteins independently of the canonical E1-E2 cascade. Here we engineer the SidE ligases to develop a modular proximity ligation approach for the identification of targets of small molecules and proteins, which we call SidBait. We validate the method with known small molecule-protein interactions and use it to identify CaMKII as an off-target interactor of the breast cancer drug ribociclib. Structural analysis and activity assays confirm that ribociclib binds the CaMKII active site and inhibits its activity. We further customize SidBait to identify protein-protein interactions and discover the F-actin capping protein (CapZ) as a target of the *Legionella* effector RavB during infection. Structural and biochemical studies indicate that RavB allosterically binds CapZ and decaps actin, thus functionally mimicking eukaryotic CapZ interacting proteins. Collectively, our results establish SidBait as a reliable tool for identifying targets of small molecules and proteins.

**Keywords** Actin Capping; Kinase Inhibitor; *Legionella*; Target Identification
**Subject Categories** Methods & Resources; Post-translational Modifications & Proteolysis

## Introduction

*Legionella pneumophila* is an intracellular bacterial pathogen that translocates over 330 effector proteins that disrupt essential cellular processes (Isberg et al, 2009), including ubiquitination (Bhogaraju et al, 2016b; Qiu et al, 2016; Zhang et al, 2021) and cytoskeleton remodeling (Zhang et al, 2023), to create a specialized replicative niche called the *Legionella*-containing vacuole (LCV). The SidE family of effectors (SdeA, SdeB, SdeC and SidE; hereafter referred to collectively as "SidE") catalyze the ligation of ubiquitin (Ub) to Ser and Tyr residues on host proteins independently of the cellular E1 and E2 Ub-conjugating enzymes (Bhogaraju et al, 2016a; Kotewicz et al, 2017; Qiu et al, 2016). The SidE effectors, which are spatially and temporally regulated during *Legionella* infection (Bhogaraju et al, 2019; Black et al, 2019; Gan et al, 2019; Shin et al, 2020; Sulpizio et al, 2019; Wan et al, 2019), ubiquitinate several host proteins to protect the LCV from degradation by the host (Kim and Isberg, 2023). This process involves the SidE ADP-ribosyltransferase (ART) domain, which transfers ADP-ribose from NAD+ to Arg42 on Ub. Subsequently, the SidE phosphodiesterase (PDE) domain hydrolyzes the phosphoanhydride bond in ADP-ribose and thereby attaches Ub to Ser or Tyr residues on host proteins through a phosphoribose (pR) linkage (Fig. 1A,B). Phosphoribosyl ubiquitination of proteins by the SidE ligases does not use the Ub C-terminal diglycine motif, nor does it require any Lys residues present in Ub or SidE substrates (Qiu et al, 2016). This orthogonality to conventional ubiquitination in the mammalian cell led us to envision a chemical proteomic approach for target identification by means of SidE-catalyzed protein capture.

Common strategies for target identification include diazirine photochemistry to cross-link ligands to proteins (Mackinnon and Taunton, 2009), engineered enzyme fusions that generate diffusible reactive species (Rhee et al, 2013; Roux et al, 2012; Tao et al, 2023), modified ubiquitin-like proteins (Hill et al, 2016; O'Connor et al, 2015) and others (Schenone et al, 2013). Each approach has drawbacks that present challenges in target validation; thus, the pursuit of complementary methods to uncover the mechanisms of action of small molecules and the substrates of enzymes remains a major objective. Here we develop a modular proximity ligation method, called SidBait, that utilizes the SidE

[1]Department of Molecular Biology, University of Texas Southwestern Medical Center, Dallas, TX 75390, USA. [2]Howard Hughes Medical Institute, University of Texas Southwestern Medical Center, Dallas, TX 75390, USA. [3]Department of Internal Medicine, University of Texas Southwestern Medical Center, Dallas, TX 75390, USA. [4]Department of Biochemistry, University of Texas Southwestern Medical Center, Dallas, TX 75390, USA. [5]Department of Microbiology, University of Texas Southwestern Medical Center, Dallas, TX 75390, USA. [6]Department of Biochemistry and Molecular Biology, University of Massachusetts Amherst, Amherst, MA 01003, USA. [7]Department of Veterinary and Animal Sciences, University of Massachusetts Amherst, Amherst, MA 01003, USA. [8]Department of Biophysics, University of Texas Southwestern Medical Center, Dallas, TX 75390, USA. [9]Harold C. Simmons Comprehensive Cancer Center, University of Texas Southwestern Medical Center, Dallas, TX 75390, USA. [10]Hamon Center for Regenerative Science and Medicine, University of Texas Southwestern Medical Center, Dallas, TX 75390, USA. [11]These authors contributed equally: James S Ye, Abir Majumdar. ✉E-mail: vincent.tagliabracci@utsouthwestern.edu

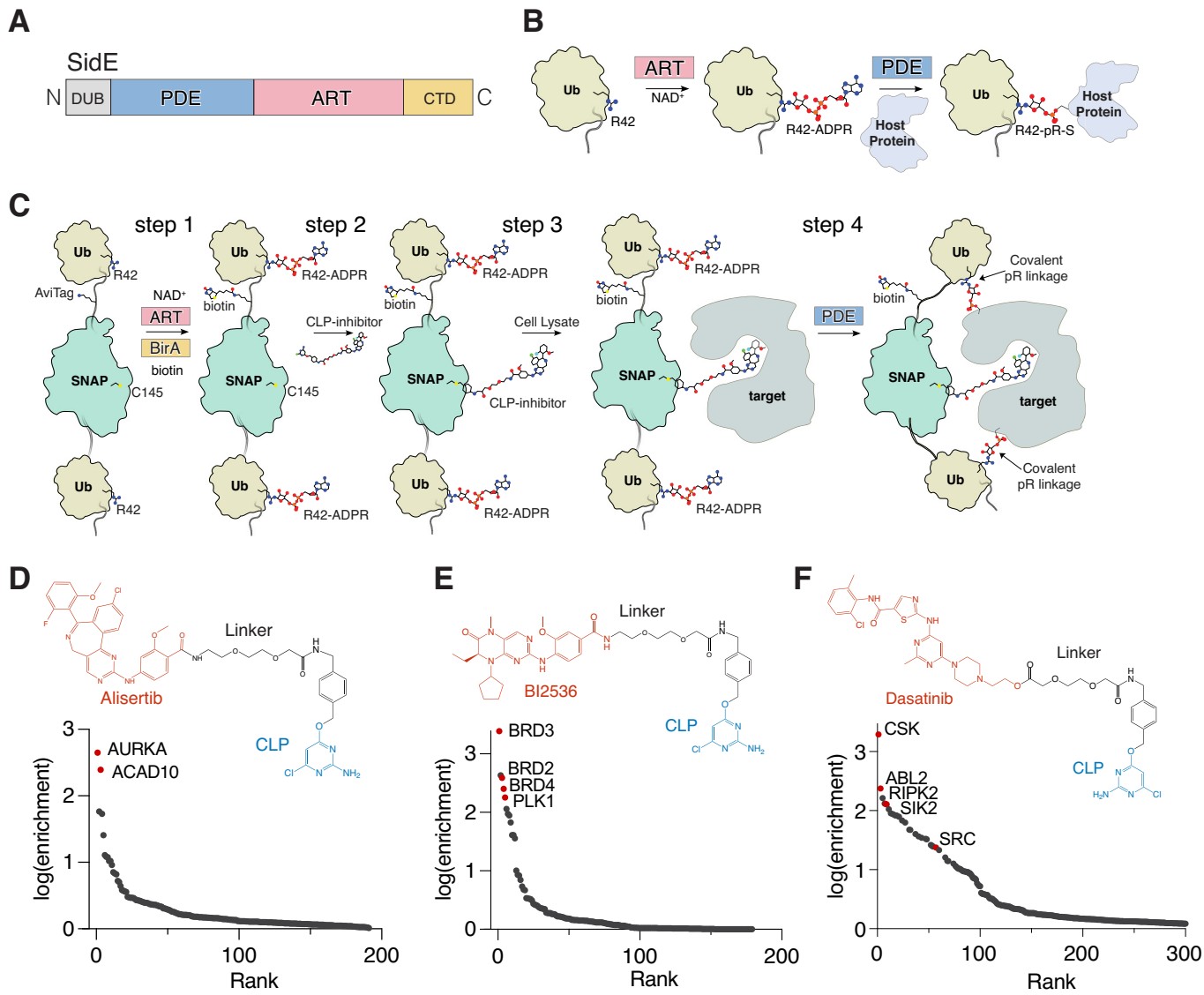

**Figure 1.   The SidBait approach to discover protein targets of small molecules.**

(**A**) Domain architecture of the SidE ligases depicting the phosphodiesterase (PDE; light blue) domain and ADP-ribosyltransferase domain (ART; pink). The deubiquitinase (DUB) and C-terminal domain (CTD) are also shown. (**B**) The enzymatic reaction catalyzed by the SidE ligases. In the first step of the reaction, the ART domain ADP-ribosylates Arg42 of Ub using $NAD^+$ as a co-substrate. Subsequently, the PDE domain hydrolyzes the phosphodiester bond and attaches the Ub to Ser and Tyr residues on host proteins. (**C**) Overall schematic of the SidBait technology. In step 1, a 6xHis-Ub-Avi-SNAP-Ub fusion protein is ADP-ribosylated by the SidE ART domain and biotinylated by BirA. In step 2, a chloropyrimidine (CLP) derivative of a small molecule is conjugated to the SNAP tag. In step 3, the probe is incubated with a cell lysate and the small molecule engages its target. In step 4, the SidE PDE domain is then added to covalently cross-link proximal interacting proteins through a phosphoribose (pR) linkage to the bait. Interacting proteins are then affinity purified by avidin pulldown and subjected to LC-MS/MS for identification. (**D–F**) Plots of fold enrichment of proteins from SidBait-alisertib (**D**), SidBait-BI2536 (**E**), and SidBait-dasatinib (**F**) experiments over pulldowns with the SidBait[C145A] control, which cannot accommodate the CLP probe. Known direct interactors are colored in red. Fold enrichment values are averaged from three independent experiments. Chemical structures of the CLP-tagged alisertib (**D**), BI2536 (**E**) and dasatinib (**F**) derivatives are shown above the plots.

ligase to covalently capture proximal interacting targets of a small molecule or a protein of interest. We showcase SidBait's utility by identifying the known targets of small molecules and proteins, thereby confirming its potential for precise target identification. We use SidBait to discover an off-target of the cyclin-dependent kinase 4/6 (CDK4/6) inhibitor ribociclib and elucidate the function of the uncharacterized *Legionella* effector RavB.

# Results

## Development of SidBait for the identification of targets of small molecules

The SidE ligases do not require a strong consensus sequence surrounding the Ser/Tyr residue for phosphoribosyl ubiquitination of substrates, resulting in low substrate specificity in vitro and when

overexpressed in cells (Kalayil et al, 2018; Zhang et al, 2021). While the ART domain specifically ADP-ribosylates Arg42 of Ub, the PDE domain is relatively indiscriminate in its substrate preferences. Importantly, the ART and PDE activities of SidE can be decoupled; for example, ADP-ribosylated Ub can be used as a substrate for the isolated PDE domain in the absence of the ART domain (Black et al, 2019).

We envisioned a strategy to exploit the nonspecificity of the SidE effectors to identify proximal interacting proteins of small molecules. We generated a bacterial expression construct containing a 6xHis tag to aid in purification, an avidin-tag for target enrichment and a SNAP-tag, which can covalently conjugate a small molecule containing a chloropyrimidine (CLP) moiety (Keppler et al, 2003), flanked by a pair of Ubs (Fig. 1C). We also mutated the diglycine motif in Ub to dialanine to prevent recognition by the endogenous eukaryotic ubiquitination machinery. We co-expressed the fusion protein with the untagged ART domain of SidE (SdeA$^{519-1100}$) and the biotin ligase BirA in E. coli and purified the fusion protein by Ni-NTA affinity chromatography (Fig. 1C; step 1). Intact mass spectrometry analysis revealed full incorporation of 2 ADPR molecules and 1 biotin (Fig. EV1a). The fully modified protein is hereafter referred to as SidBait.

We incubated SidBait with SidE and observed NAD$^+$-independent autoubiquitination, indicated by the high molecular weight crosslinks, thus confirming that the ADP-ribosylated Ubs serve as substrates for the SidE PDE domain (Fig. EV1B). To attach a small molecule to the bait, we synthesized a CLP derivative of the Aurora kinase A (AURKA) inhibitor alisertib (Fig. 1D; top), which inhibits AURKA with a similar IC$_{50}$ to the unmodified drug (Bucko et al, 2019; Manfredi et al, 2011). We then incubated CLP-alisertib with SidBait and observed full occupancy of the compound into the SNAP-tag (SidBait-alisertib), but not a SNAP$^{C145A}$ mutant, which is incapable of incorporating the CLP tag (Figs. 1C; step 2, EV1C,D).

To determine whether SidBait-alisertib could enrich AURKA, we incubated the probe with a HEK293 cell lysate (Fig. 1C; step 3), added the SidE PDE domain (SdeA$^{178-1100}$) to cross-link any proximal interacting proteins (Fig. 1C; step 4) and enriched the bait by avidin pulldown. As a control, we used SidBait containing the SNAP$^{C145A}$ mutant. Mass spectrometry analysis identified AURKA and the known off-target alisertib interactor ACAD10 (Adhikari et al, 2020) as the most highly enriched proteins (Fig. 1D, Dataset EV1). Free alisertib outcompeted SidBait-alisertib for AURKA, confirming that the target binding was solely due to the SidBait-alisertib probe (Fig. EV1E). These experiments also revealed that AURKA was completely depleted from the cell lysate by the probe.

As further validation for this approach, we synthesized CLP derivatives of the dual PLK1 and BRD4 inhibitor BI2536 (Ciceri et al, 2014), and the SRC-family tyrosine kinase inhibitor dasatinib (Lombardo et al, 2004) (Fig. 1E,F; top), and performed SidBait experiments. Mass spectrometry analysis identified the expected targets of these drugs, and SidBait experiments conducted with K562 cell lysates produced similar results (Figs. 1E,F and EV2; Datasets EV2,3). Thus, SidBait identifies protein targets of small molecules.

## SidBait identifies CaMKII as a target of ribociclib

We next performed SidBait using the clinically approved CDK4/6 inhibitor ribociclib, which is used in the treatment of breast cancer (Hortobagyi et al, 2022). Although we initially failed to detect

CDK4/6, we unexpectedly identified calcium/calmodulin-dependent protein kinase type II (CaMKII) paralogs as top hits in HEK293, human cardiomyocyte, and MCF7 cell lysates (Figs. 2A and EV3A,B; Dataset EV4). However, incubation of SidBait-ribociclib in a cell lysate for 72 h prior to the addition of the SidE PDE domain resulted in the enrichment of CDK4 (Fig. EV3C, Dataset EV4), demonstrating that incubation time prior to cross-linking is a variable that can be strategically optimized.

Consistent with the SidBait experiments, ribociclib inhibited CaMKIIδ activity in vitro and endogenous CaMKII in cultured cardiomyocytes (Figs. 2B,C and EV3D). Moreover, ribociclib-treated cardiomyocytes showed a significant decrease in beating rate, a known effect of CaMKII inhibition (Wu et al, 2009) (Fig. 2D). We determined the crystal structure of the kinase domain of CaMKIIδ (CaMKIIδ$^{11-309}$; hereafter referred to as CaMKIIδ$^{kd}$) bound to ribociclib (Figs. 2E and EV3E; Appendix Table S1). Ribociclib occupies the active site of CaMKII in a manner analogous to CDK6 (Chen et al, 2016), with the dimethylcarboxamide group of ribociclib packing against the gatekeeper Phe90 (Fig. 2F,G). There are hydrogen bonding interactions within the CaMKIIδ$^{kd}$ hinge region and with the backbone nitrogen of Asp157. Residues Glu97 and Glu100, which are on the αD-helix of CaMKII and coordinate substrates and the ribose hydroxyls of ATP (Yang and Schulman, 1999) (Fig. 2F), provide a solvent-exposed ridge, like the one in CDK6 (Asp104 and Thr107), to stabilize the piperazine ring of ribociclib (Chen et al, 2016) (Fig. 2G). The lower potency of ribociclib for CaMKII compared to CDK4/6 is likely to arise from the presence of a leucine (CaMKII Leu92) in place of the histidine residue (CDK6 His100) that imparts selectivity for ribociclib (Cho et al, 2010) (Figs. 2B and EV3F–H). Collectively, our structural analysis shows that CaMKIIδ readily accommodates ribociclib and may facilitate the development of more selective CaMKII and CDK4/6 inhibitors.

## Development of SidBait for the identification of protein targets

Replacing the SNAP-tag within SidBait with a protein of interest could, in principle, also be used to identify protein-protein interactions. To explore the modularity of SidBait, we replaced the SNAP-tag with: the Aurora A-activating segment of Tpx2 (Tpx2$^{1-43}$); HopBF1, a bacterial effector kinase that phosphorylates eukaryotic HSP90 (Lopez et al, 2019); and p38β, a member of the mitogen activated protein kinase (MAPK) family (MAPK11). We co-expressed the fusion proteins with the SidE ART domain and BirA in E. coli (Fig. 3A; step 1). Intact mass analysis revealed full incorporation of 2 ADPR molecules and 1 biotin (Fig. EV4A,B). We incubated these proteins with SidE and observed NAD$^+$-independent autoubiquitination, thus confirming that the ADP-ribosylated Ubs serve as substrates for the SidE PDE domain (Fig. EV4C). We then incubated SidBait-Tpx2$^{1-43}$, SidBait-HopBF1 and SidBait-p38β with HEK293 cell lysates (Fig. 3A; step 2), added the SidE PDE domain to cross-link any proximal interacting proteins (Fig. 3A; step 3) and enriched the baits by avidin pulldown. Mass spectrometry analysis identified AURKA as highly enriched in the SidBait-Tpx2$^{1-43}$ experiment (Fig. 3B, Dataset EV5), and HSP90 in the SidBait-HopBF1 experiment (Fig. 3C, Dataset EV6). SidBait-p38β identified several interactors of p38β, including direct substrates and the upstream kinase MAP2K4 (Ho et al, 2003) (Fig. 3D,

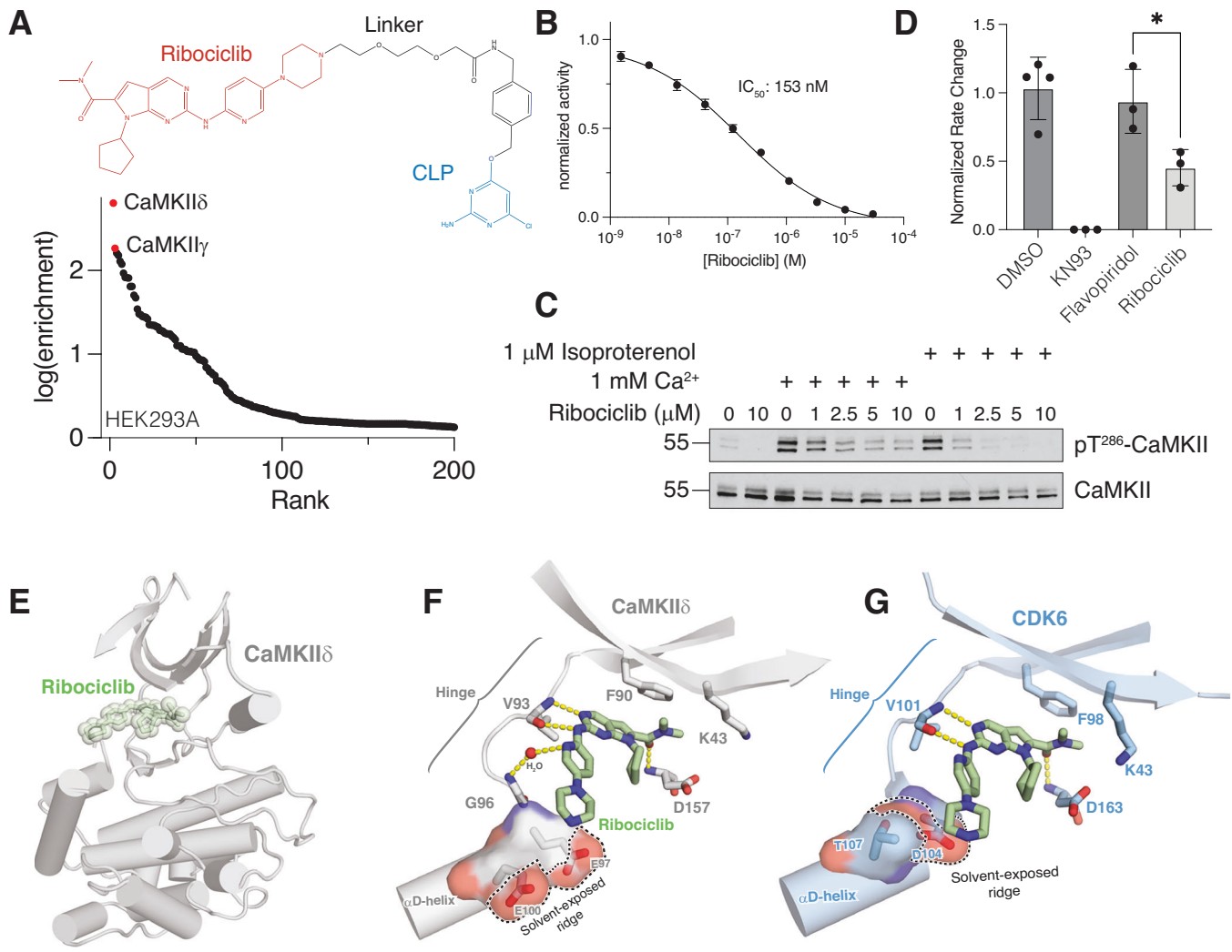

**Figure 2. SidBait identifies ribociclib as an active site inhibitor of CaMKII.**

(A) Plot of fold enrichment of proteins from SidBait-ribociclib experiments in HEK293A cells over the SidBait[C145A] control. Fold enrichment values are averaged from three independent experiments. The chemical structures of CLP-tagged ribociclib is shown above the plot. (B) In vitro CaMKIIδ activity assay against a peptide substrate in the presence of varying concentrations of ribociclib. The IC$_{50}$ of ribociclib is shown in the inset. Error bars represent the S.E.M. of three replicates. (C) Protein immunoblot of human cardiomyocyte lysates that have been stimulated with Ca$^{2+}$ or isoproterenol in the presence of varying concentrations of ribociclib. Total CaMKII is shown to demonstrate steady levels of CaMKII; pThr$^{286}$ CaMKII is shown as an indication of endogenous CaMKII activity. (D) Fold change in beating rate of cardiomyocytes following treatment with the CaMKII inhibitor KN93, the pan-CDK inhibitor flavopiridol or the CDK4/6 inhibitor ribociclib. Results represent the mean of three independent experiments. Error bars represent standard deviation. (E) Overall structure of the CaMKIIδ kinase domain (gray) with ribociclib (green spheres) bound in the active site. (F, G) Zoomed in view of the active sites of CaMKII in gray (F) and CDK6 in light blue (G; PDB: 5L2T) depicting the interactions between the kinase domain and ribociclib (green). Source data are available online for this figure.

Dataset EV7). Thus, SidBait can capture protein-protein interactions, including potential substrates of enzymes and enzymes for substrates.

## The *Legionella* effector RavB targets the F-actin capping complex CapZ

The identification of host targets of bacterial effectors is a significant challenge in host-pathogen biology. To test whether SidBait can be applied to these questions, we performed SidBait with the uncharacterized *Legionella* effector RavB (lpg0030). Analysis of the AlphaFold structural prediction suggests that RavB

contains a helical N-terminal segment and a lipid binding C-terminal domain connected by a long unstructured linker (Fig. 4A) (Jumper et al, 2021). SidBait-RavB identified both subunits (CapZα and CapZβ) of the heterodimeric F-actin capping complex (CapZ) (Fig. 4B, Dataset EV8) as the target of RavB. Free RavB outcompeted SidBait-RavB for CapZ, confirming that the target binding was due solely to the presence of RavB (Fig. EV5A,B). CapZ binds to the barbed ends of actin filaments (F-actin) and regulates actin dynamics and the phagocytosis of bacteria (Edwards et al, 2014). RavB and CapZ formed a stable heterotrimeric complex with a $K_d$ of 10.8 nM (Fig. 4C).

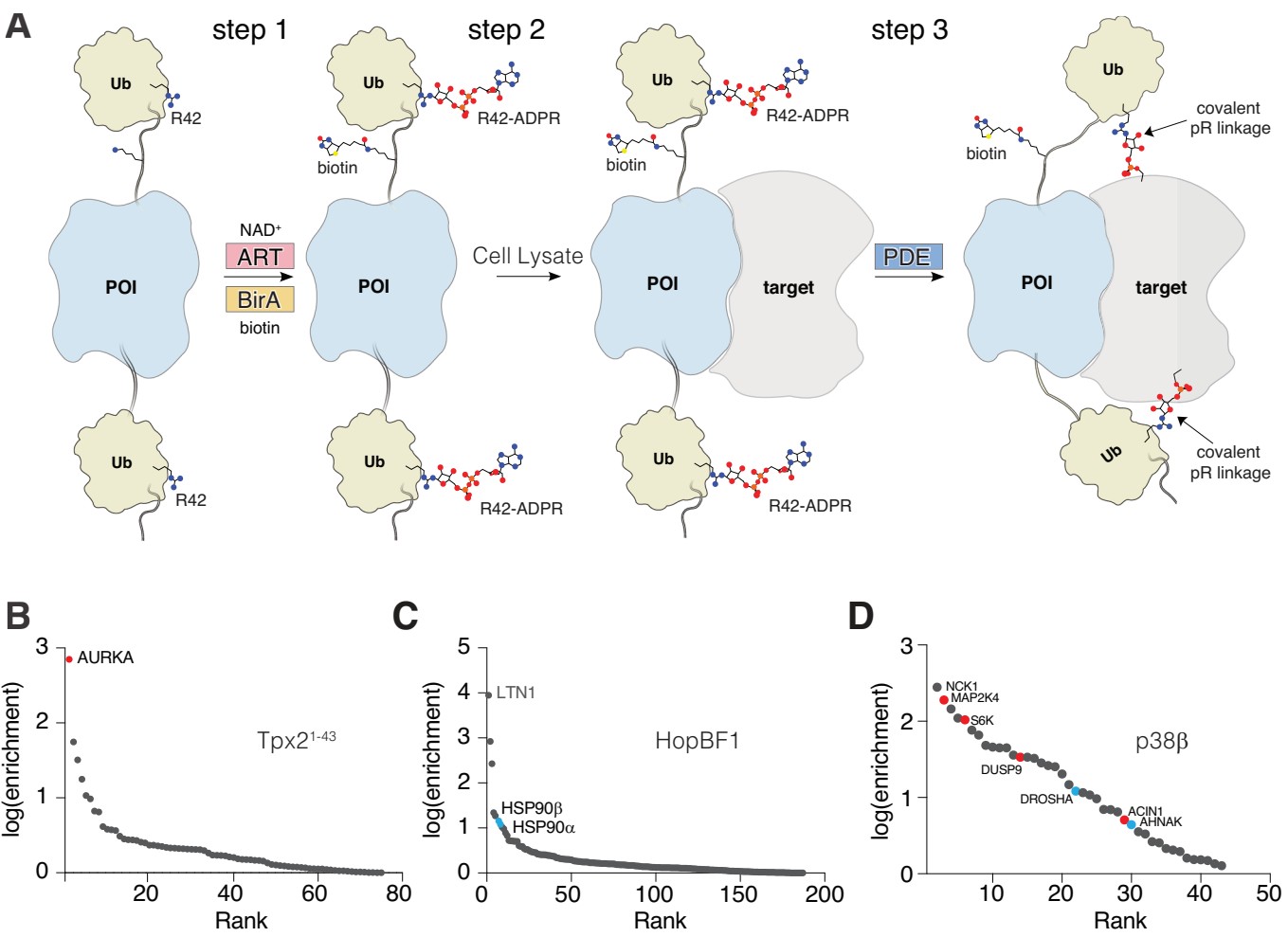

**Figure 3. The SidBait approach to discover targets of proteins of interest.**

(A) Overall schematic of the SidBait approach with a protein of interest (POI). In step 1, a SidBait molecule in which the SNAP-tag is replaced with a POI is ADP-ribosylated by the SidE ART domain and biotinylated by BirA. In step 2, the probe is incubated with a cell lysate and the POI engages its binding partners. In step 3, the SidE PDE domain is then added to covalently cross-link proximal interacting proteins through a pR linkage to the bait. Interacting proteins are then affinity purified by avidin pulldown and subjected to LC-MS/MS for identification. (B–D) Plots of fold enrichment of proteins from SidBait-Tpx2[1-43] (B), SidBait-HopBF1 (C) and SidBait-p38β (D). Fold enrichment values are averaged over three independent experiments. Known interacting partners are in red; known substrates are in blue.

We solved the structure of the RavB:CapZ complex to a resolution of 2.00 Å. The structure resolved CapZ bound to an internal segment of RavB[115-139] (Fig. 4D; Appendix Table S1). CapZ consists of a heterodimer of CapZα and CapZβ, which produce a mushroom shaped structure with an α-helical stalk and an F-actin binding cap. RavB[115-139] wraps around the stalk of the CapZ mushroom distal to the F-actin binding cap. RavB[115-139] contains a highly conserved motif, [115]L(X)$_6$R(X)$_6$RRLP[132] (Fig. 4E) (hereafter referred to as RavB Capping protein interacting motif; RavB[CPI]), in which the arginines form electrostatic interactions with Asp44, Asp63, and Asp85 of CapZβ (Fig. 4F). Alanine substitutions of Arg122, Arg129 and Arg130 (RavB[3R3A]) abolished RavB binding to CapZ in vitro and in cells (Figs. 4G and EV5C). Likewise, a synthesized RavB[108-148] peptide was able to bind CapZ with high affinity (Fig. EV5D). Thus, RavB[CPI] is necessary and sufficient for its interaction with CapZ.

RavB also contains a C-terminal phosphatidylinositol phosphate (PIP) binding domain (RavB[CTD]) (Nachmias et al, 2019) (Fig. 4a).

In HeLa cells, EGFP-tagged RavB recruited CapZ to LAMP-1 positive vesicular structures, suggesting ectopically expressed RavB exhibits lysosomal localization (Fig. 5A,B and EV6A,B). RavB[3R3A] and RavB[CTD] also localized to the lysosome, but were unable to recruit CapZ. RavB and RavB[3R3A] bound PI3P, PI4P, and PI(3,5)P$_2$, while the H214A mutant––predicted to disrupt PIP binding––was markedly impaired (Figs. 5C and EV6C). Likewise, RavB[H214A] failed to localize to the lysosome when expressed in cells (Figs. 5A,B and EV6A,B). In *Legionella* infected cells, GFP-RavB[H214A] but not RavB[3R3A/H214A] co-localized with CapZ on the LCV (Fig. 5d).

The CapZ interacting proteins, CD2AP, CARMIL1 and CKIP-1, decap actin and contain membrane targeting domains used to spatially regulate actin dynamics (Edwards et al, 2014). These proteins also bind in the same allosteric groove as RavB using a similar CPI motif (Bruck et al, 2006) (Fig. 5E,F). Indeed, RavB, but not RavB[3R3A], displaced CapZ from F-actin (Fig. 5G,H). Furthermore, RavB[CPI] and the CapZ binding region from CARMIL1

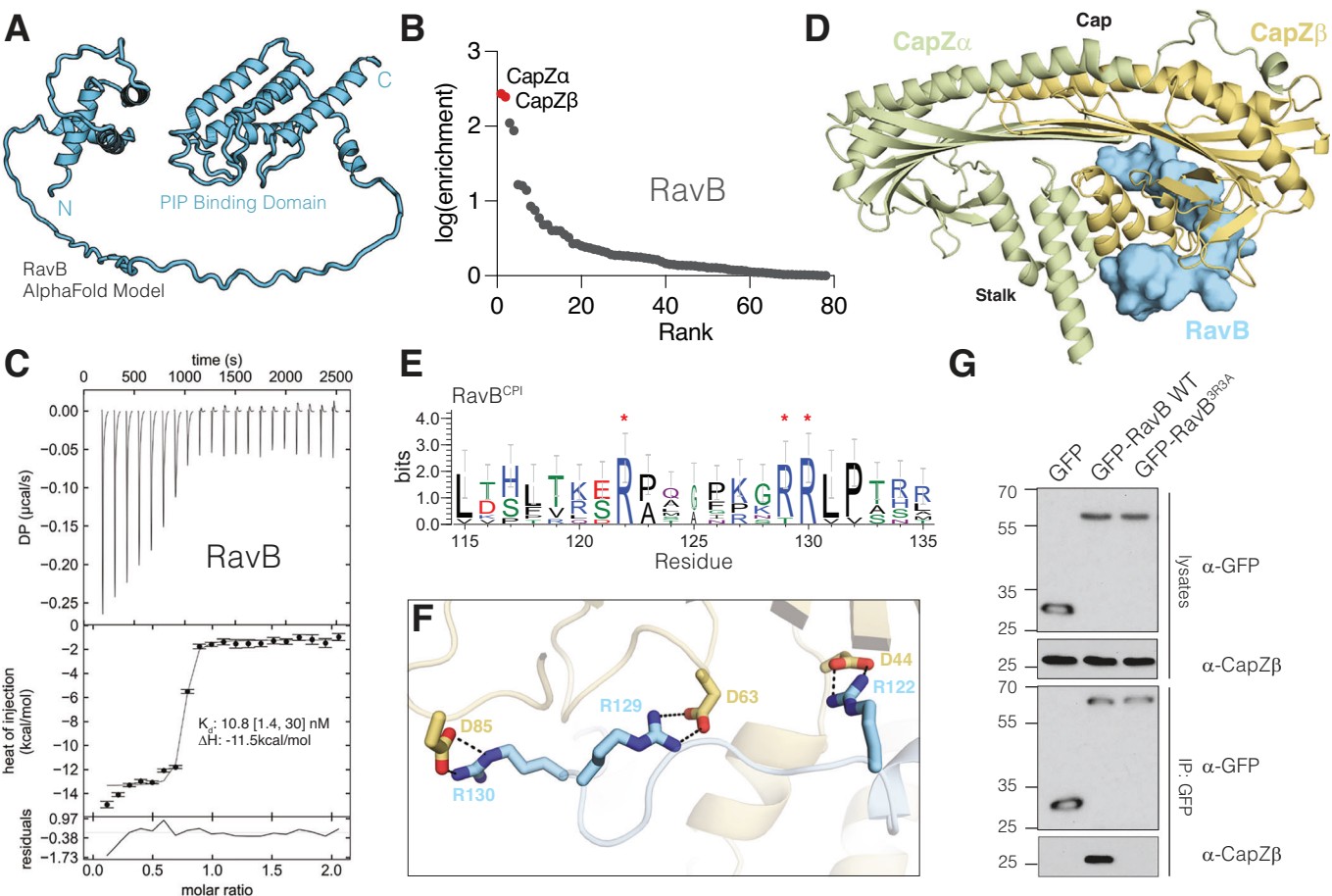

**Figure 4. SidBait identifies CapZ as a binding partner of the *Legionella* effector RavB.**

(A) Structural model of RavB predicted by AlphaFold2. (B) Plot of fold enrichment of proteins from SidBait-RavB. Fold enrichment values are averaged over three independent experiments. (C) Isothermal titration calorimetry (ITC) trace showing the binding of RavB (injected) to CapZ (cell). Error bars represent the standard error from the integration of the injection peaks of the thermogram. $K_d$ and enthalpy values are shown in the inset with a 68.3% confidence interval ($n = 1$). (D) Overall structure of RavB (cyan) bound to CapZα (green) and CapZβ (yellow). The F-actin binding region ("Cap") and the RavB binding region ("Stalk") are labeled. (E) Sequence logo representing conservation of the RavB^CPI compiled across 45 homologs. Conserved arginine residues that directly participate in electrostatic interactions with CapZ are annotated with red asterisks. (F) Zoomed in view depicting the direct electrostatic interactions between RavB^CPI (cyan) and CapZβ stalk (yellow). (G) Protein immunoblot of lysates and anti-GFP-immunoprecipitates from HEK293 cells that have been transfected with GFP, GFP-RavB or the GFP-RavB^3R3A. anti-GFP and anti-CapZβ immunoblots are shown. Source data are available online for this figure.

(CBR115) (Hernandez-Valladares et al, 2010) decapped actin and promoted its polymerization (Figs. 5I and EV6D). Collectively, our results suggest that RavB functionally mimics eukaryotic CapZ interacting proteins by allosterically binding CapZ and decapping F-actin.

## Expanding the applications of SidBait

A key advantage of SidBait is that the technique is agnostic to the source of target proteins. Using baits we had already tested against cell lysates, we expanded our protein sources to include yeast and animal tissues. We performed SidBait experiments using ribociclib and dasatinib in a homogenized mouse heart and kidney, respectively. As expected, SidBait-ribociclib resulted in an enrichment of CaMKIIδ and γ (Fig. 6A, Dataset EV4), while SidBait-dasatinib yielded the tyrosine kinases Csk, Lyn, Yes and Src (Fig. 6B, Dataset EV3). From yeast extracts, SidBait experiments

using the bacterial effector AnkD resulted in the enrichment of its expected target USO1 (Chen et al, 2022) (Fig. 6C, Dataset EV9).

To evaluate the in vivo applicability of SidBait, we serially injected various baits and the SidE PDE protein into live *X. laevis* embryos. Using SidBait-ribociclib, SidBait-p38β, and SidBait-dasatinib, we successfully captured many of the expected targets (Figs. 6D,E and EV7A, Datasets EV3, 4, 7), thus establishing the method's functionality in a live system.

A limitation of SidBait is that bait proteins must be expressed and ADP-ribosylated in *E. coli* because SidE-dependent ADP-ribosylation of endogenous ubiquitin in eukaryotic systems causes cytotoxicity (Bhogaraju et al, 2016a). To circumvent this issue, we used the SpyTag-SpyCatcher system (Keeble et al, 2019), which enables the fusion of two separate polypeptides via an isopeptide bond (Fig. EV7B). As a proof-of-concept, we purified the Ub-SpyCatcher-Ub and SpyTag-RavB proteins separately, conjugated them and performed a SidBait experiment from a cell lysate. As

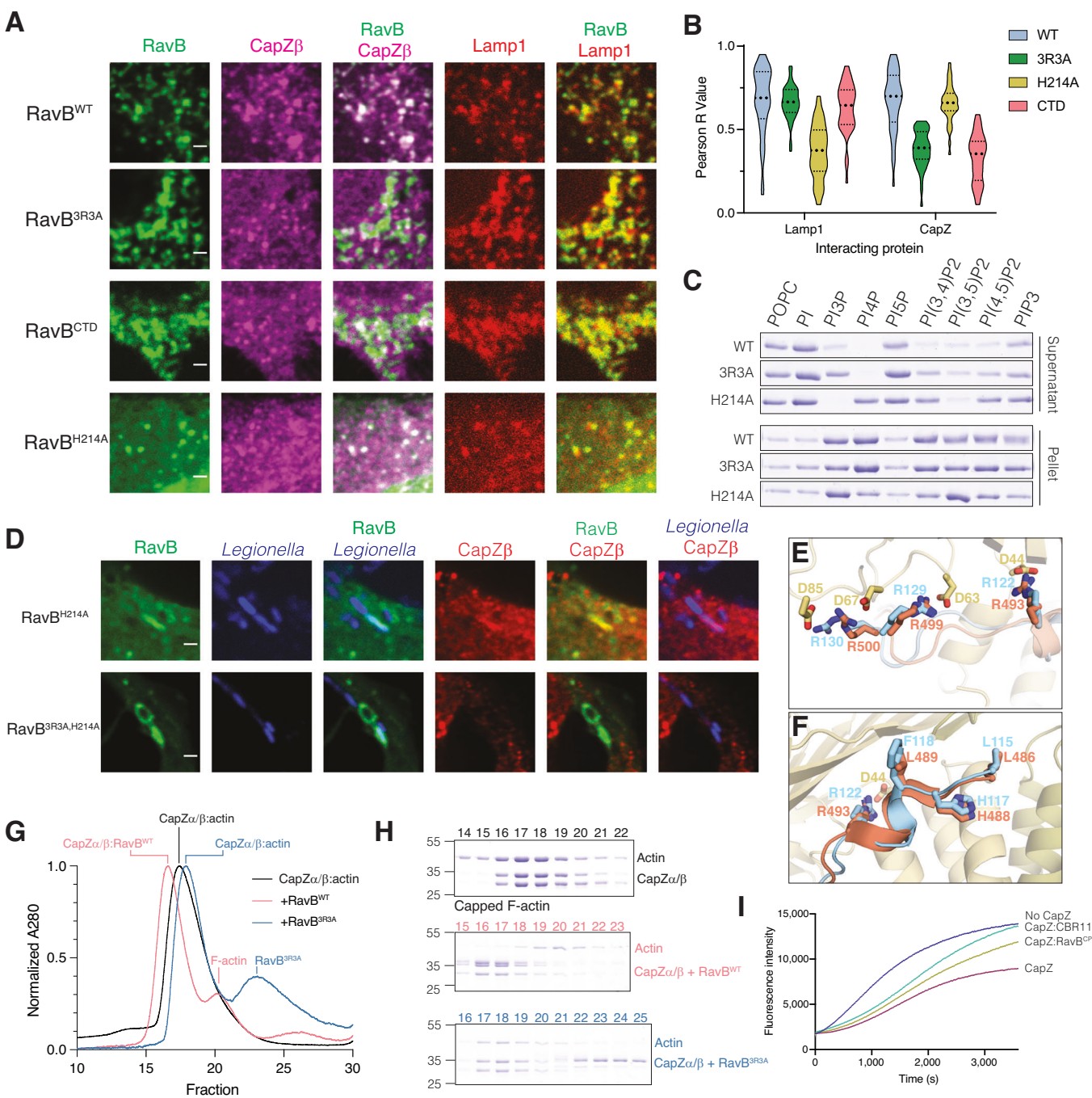

**Figure 5. RavB acts as a lipid bound F-actin decapping protein.**

(A) Immunofluorescence microscopy of HeLa cells expressing EGFP-RavB[WT] or various mutants, mTagBFP-Lamp1 and endogenous CapZβ. Scale bar represents 2 μm. (B) Plot depicting Pearson R Value of colocalization between transfected GFP-RavB, or various RavB mutants with mTagBFP-Lamp1 and endogenous CapZβ. Each comparison was calculated using 40 cells across 3 independent experiments. (C) Liposome sedimentation assay with different phospholipids and RavB or mutants. RavB bound liposomes were separated into supernatant (unbound) or pellet (bound) by high-speed centrifugation and RavB was visualized by Coomassie staining. (D) Immunofluorescence microscopy of mCherry-*Legionella*-infected (pseudocolored in blue) COS-7 cells expressing EGFP-RavB[H214A] or EGFP-RavB[3R3A/H214A]. Endogenous CapZβ was also visualized (red). Scale bar represents 2 μm. (E, F) Zoomed in view of CapZ (yellow):RavB (cyan) structure with human CD2AP (orange, PDB: 3LK4) overlayed depicting direct electrostatic interactions (E) or the N-terminal segment of the RavB and CD2AP CPIs (F). (G) Size exclusion chromatography depicting the A280 traces from the actin decapping experiments. CapZ-capped F-actin is shown in black, purified CapZ-capped F-actin incubated with RavB[WT] is in red, and CapZ with the RavB[3R3A] mutant in green. (H) Proteins present in fractions from size exclusion runs in (G) were visualized by Coomassie staining. (I) Pyrene-actin polymerization assays demonstrating that the RavB[CPI] decaps actin. The polymerization of actin was measured in the presence of a buffer control (blue), CapZ (maroon), CapZ and the known decapping peptide CBR115 (teal), and CapZ and the Rav[CPI] peptide (green). Assays were performed with 2 μM G-actin, 50 nM CapZ and 1.25 μM decapping peptide. Source data are available online for this figure.

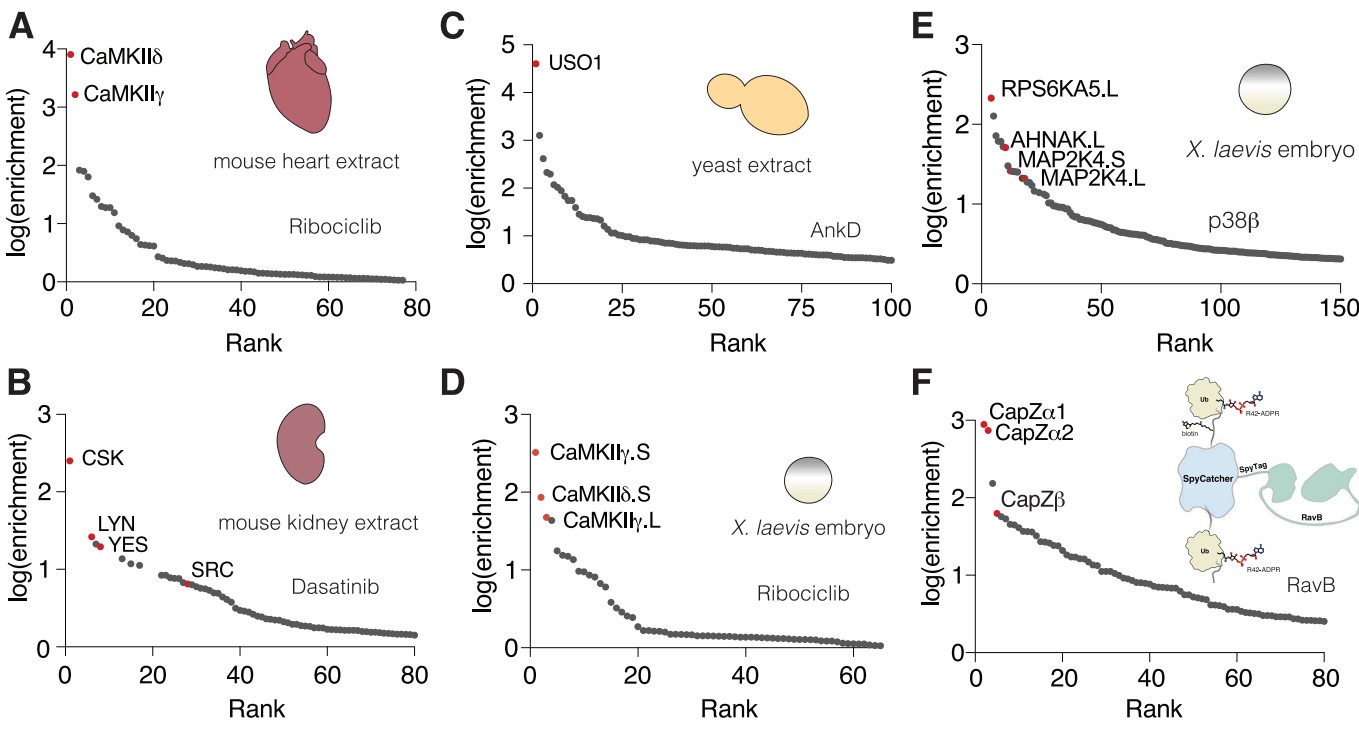

**Figure 6. Additional applications of SidBait.**

(A, B) Plots of fold enrichment of proteins from SidBait-ribociclib in a mouse heart extract (A) and SidBait-dasatinib from a mouse kidney extract (B) over pulldowns with the SidBait$^{C145A}$ control. (C) Plot of fold enrichment of proteins from SidBait-AnkD from yeast extracts over pulldowns with the SidBait protein control. (D, E) Plots of fold enrichment of proteins from SidBait-ribociclib (D) and SidBait-p38β (E) from live *X. laevis* embryos over pulldowns with the respective controls. (F) Plot of fold enrichment of proteins from a SpyTag/SpyCatcher fusion of RavB over pulldowns with the non-conjugated control. Known interactors are shown in red.

expected, the CapZ complex was highly enriched (Fig. 6F, Dataset EV8). We envision this approach to be particularly useful for applications that require alternative expression systems for bait proteins.

## Discussion

In this work we introduce SidBait, a proximity ligation approach that complements current methodologies to identify targets of small molecules and proteins. We validate the method by recapitulating known bait-target pairs and demonstrate its utility as a hypothesis-generator by identifying and further substantiating previously unexplored interactions. These discoveries were facilitated by key features of SidBait that distinguish it from existing proximity ligation techniques: modularity, which allows for efficient and reliable target identification of both small molecules and proteins using the same method; a covalent linkage between the bait and target, which permits stringent washing to reduce background and false positives; and widely tunable experimental parameters, such as temperature (4–37 °C), incubation time and concentrations of bait and lysate.

For example, our initial failure to detect CDK4 with ribociclib (Fig. 2A) may have been due to the inability of CDK4 to bind ATP-competitive inhibitors in its heterotrimeric CDK4:CyclinD:p27 form (Guiley et al, 2019). This led us to attempt a significantly

longer incubation with SidBait-ribociclib than proximity ligation approaches typically permit, thus allowing more time to capture a different form of CDK4 that exists in equilibrium (Fig. EV3C). It is also likely that the relative cellular abundances of proteins influenced the results, potentially explaining the enrichment of CDK6 instead of CDK4 after treatment with SidBait-ribociclib in K562 cell lysates (Fig. EV2). Meanwhile, our unexpected result showing ribociclib to be a potent CaMKII inhibitor are consistent with reports that have noted this interaction in passing (Chen et al, 2016; Kim et al, 2018; Sumi et al, 2015). Interestingly, some breast cancer patients develop serious cardiovascular side effects upon treatment with ribociclib, including long QT syndrome (Barber et al, 2019). Because CaMKII responds to calcium flux in cardiomyocytes and regulates cardiac pacemaking (Bers and Grandi, 2009), we speculate that ribociclib targeting CaMKII may contribute to the observed heart toxicity in a small subset of patients (Infante et al, 2016).

While SidBait is effective when injected into frog embryos (Fig. 6D,E), its ability to be used in lysates, particularly for small molecule target identification, is a notable strength of the technology. For instance, our results with SidBait-alisertib using HEK293 cell lysates unambiguously identified the intended target, Aurora A, as well as ACAD10, an off-target interactor (Adhikari et al, 2020; Klaeger et al, 2017), as the two most highly enriched targets. In contrast, genetically encoded proximity ligation methods that also used alisertib from HEK293 cells failed to identify

ACAD10 (Tao et al, 2023), which is a mitochondrial protein (Rashan et al, 2025); genetically encoded systems are unlikely to penetrate the mitochondria unless programmed to do so. Thus, SidBait is not limited to any specific subcellular compartment as it effectively interacts with all proteins within the lysate—a particularly valuable feature when identifying off-targets of small molecules.

In two other small molecule target identification studies performed in cells with dasatinib, the authors report interactors using photocatalysis-mediated cross-linking (Trowbridge et al, 2022) and proximity-based compound-binding protein identification (Kwak et al, 2022). Unlike these methods, we successfully identified SIK2, a known dasatinib interactor (Montenegro et al, 2020; Ozanne et al, 2015) that is not a Src-family kinase. This additional sensitivity in SidBait experiments is also observed in K562 cell lysates when compared to thermal shift experiments conducted in whole cells (Fig. EV2) (Van Vranken et al, 2024).

A limitation of small molecule SidBait experiments is the requirement for a CLP derivative of each small molecule being tested. Structure-activity relationship data, should it exist, can be used to inform placement of the CLP handle. As a last resort, it may be necessary to attach the handle at multiple points of a given small molecule, but there remains a possibility that some compounds are impossible to derivatize without interfering with target binding.

Several pieces of evidence suggest that SidBait can capture transient interactions. Our data with p38β demonstrates that we can enrich the upstream kinase, MAP2K4, which interacts with p38β in the high micromolar range (Ho et al, 2003) (Fig. 3D). Furthermore, we can enrich p38β substrates, including several that, to the best of our knowledge, do not form stable complexes with p38β. Finally, the HSP90 kinase HopBF1 does not stably interact with HSP90 in vitro and has a $K_m$ of ~3 μM for HSP90 (Lopez et al, 2019). Using SidBait-HopBF1, we successfully enriched HSP90 from HEK293 cells (Fig. 3C).

The unambiguous hits from the SidBait-RavB experiment allowed us to focus on the effector's mechanism as an actin decapper. Legionella frequently exploits the host cytoskeletal network, with multiple effectors specifically targeting actin-regulating proteins or actin itself. Similarly to RavB, MavH binds CapZ and enhances actin polymerization at the LCV during phagocytosis, thus increasing bacterial uptake (Zhang et al, 2023). Likewise, LegK2 inhibits actin nucleation at the LCV to evade lysosomal fusion and bacterial destruction (Michard et al, 2015). Because CapZ is required for vesicle trafficking to lysosomes (Wang et al, 2021), we propose that RavB may be working in concert with other effectors located at the LCV to block actin- and CapZ-mediated lysosomal trafficking, thus ensuring the survival of the bacterium.

We have engineered the SidE all-in-one ubiquitin ligase to identify targets of small molecules and proteins. As with any other technique, we do not expect SidBait to work for every use case. However, we contend that between SidBait and the other proximity ligation techniques, there are overlapping but also mutually exclusive use cases. We anticipate that its versatility, orthogonal mechanism and ease of use may facilitate the discovery of important biomolecular interactions.

# Methods

**Reagents and tools table**

| Reagent/Resource | Reference or Source | Identifier or Catalog Number |
|---|---|---|
| **Experimental models** | | |
| MCF7 | ATCC | HTB-22 |
| COS-7 | ATCC | CRL-1651 |
| HeLa | ATCC | CCL-2 |
| HEK293A | J. Jewell | |
| K562 | D. Nijhawan | |
| H9 | WiCell | WA09 |
| BY4741 | Dharmacon | YSC1053 |
| Rosetta-DE3 | Novagen | 70954-3 |
| *Legionella pneumophila* Lp02 | R. Isberg | |
| **Recombinant DNA** | | |
| pET28a(+) | Novagen | 69864 |
| pETDuet-1 | Novagen | 71146 |
| pJB908 | (Sexton et al, 2005) | |
| **Antibodies** | | |
| Aurora A Rb D3E4Q | Cell Signaling | 14475S |
| p-CaMKII (Thr286) Rb D21E4 | Cell Signaling | 12716S |
| CaMKII (pan) Rb D11A10 | Cell Signaling | 4436S |
| p-phospholamban (S16/T17) | Cell Signaling | 8496S |
| CAPZα1 Ms | Proteintech | 66066-1-IG |
| CAPZβ Rb | Sigma-Aldrich | AB6017 |
| GAPDH Ms (GA1R) | Abcam | ab125247 |
| GFP Living Colors A.v. Ms (JL-8) | Takara Bio | 632381 |
| HSP90 α/β Ms (F-8) | Santa Cruz | sc-13119 |
| *Legionella pneumophila* LPS Ms | ThermoFisher Scientific | AWB4CE4 |
| Amersham ECL Rabbit IgG, HRP-linked whole Ab (from donkey) | Cytiva | NA934 |
| Amersham ECL Mouse IgG, HRP-linked whole Ab (from sheep) | Cytiva | NXA931 |
| Donkey α-Rabbit IgG (H + L) Highly Cross-Adsorbed Secondary Antibody, Alexa Fluor 647 | ThermoFisher Scientific | A-31573 |
| **Oligonucleotides and other sequence-based reagents** | | |
| See Appendix Table S2 | | |
| **Chemicals, Enzymes and other reagents** | | |
| Alisertib | MedChem Express | HY-10971 |
| Ribociclib | AdooQ BioScience | A13549 |
| Ribociclib | Selleck Chemicals | S7440 |
| KN-93 phosphate | Selleck Chemicals | S7423 |

| Reagent/Resource | Reference or Source | Identifier or Catalog Number |
|---|---|---|
| Flavopiridol | Selleck Chemicals | S1230 |
| CHIR99021 | Tocris | 4423 |
| Wnt-C59 | Cayman Chemical | 16644 |
| Isoproterenol | Sigma-Aldrich | I5752 |
| CLP-alisertib | (Bucko et al, 2019) | |
| CLP-BI2536 | This study | |
| CLP-ribociclib | WuXi Apptec | |
| CLP-dasatinib | WuXi Apptec | |
| DTT | Goldbio | DTT100 |
| TCEP HCl | Goldbio | TCEP1 |
| Kanamycin monosulfate | Goldbio | K-120-100 |
| Ampicillin (Sodium) | Goldbio | A-301-100 |
| IPTG | Goldbio | I2481C |
| PMSF | Goldbio | P-470-10 |
| D-biotin | Goldbio | D-950-100 |
| Streptavidin agarose resin | Goldbio | S-105-10 |
| Streptavidin magnetic beads | ThermoFisher Scientific | 88816 |
| HisPur Ni-NTA resin | ThermoFisher Scientific | 88222 |
| DMSO | Sigma-Aldrich | D2650-100ML |
| cOmplete, EDTA-free protease inhibitor | Roche | 5056489001 |
| DMEM | Gibco | 11965118 |
| RPMI 1640 | Gibco | 11875093 |
| Fetal bovine serum | Sigma-Aldrich | F-2442-500ML |
| Penicillin-streptomycin | Sigma-Aldrich | P0781-100ML |
| PolyJet | SignaGen | SL100688 |
| TransIT-X2 Delivery system | Mirus Bio | MIR 6000 |
| mTeSR Plus | Stemcell Technologies | 100-0276 |
| Matrigel | Corning | 354277 |
| GFP-Catcher | Antibodies-Online | ABIN5311508 |
| Actin protein | Cytoskeleton, Inc. | APHL99-C |
| Actin-pyrene polymerization kit | Cytoskeleton, Inc. | BK003 |
| PfuTurbo | Agilent | 600250 |
| dNTP mix | ThermoFisher Scientific | R0191 |
| Q5 2x Master Mix | NEB | M0492S |
| Tris base | Fisher Scientific | BP152-1 |
| HEPES | Sigma-Aldrich | H4034 |
| Imidazole | Sigma-Aldrich | I2399 |

| Reagent/Resource | Reference or Source | Identifier or Catalog Number |
|---|---|---|
| NaCl | Fisher Scientific | BP358-10 |
| LiCl | Sigma-Aldrich | L9650-100 |
| CaCl$_2$ | Sigma-Aldrich | C4901-500G |
| SDS | Sigma-Aldrich | L5750-1kg |
| EDTA | Sigma-Aldrich | E5134-500 |
| Triton-X100 | Sigma-Aldrich | T9284 |
| Tween-20 | Sigma-Aldrich | P1379-1L |
| Sodium deoxycholate | Sigma-Aldrich | D6750 |
| NP-40 alternative | Millipore | 492016 |
| PEG3350 | Sigma-Aldrich | P4338-1kg |
| 08:0 PI(3)P | Avanti Research | 850187 |
| ACES | Sigma-Aldrich | A9758-250G |
| Formaldehyde aqueous solution | Electron Microscopy Sciences | 15711 |
| Albumin | Goldbio | A-420-250 |
| PIP strip | Echelon Biosciences | P-6001 |
| SuperSignal West Pico | ThermoFisher Scientific | 34579 |
| **Software** | | |
| HKL-3000 | (Minor et al, 2006) | |
| PHASER | (McCoy et al, 2007) | |
| Buccaneer | (Cowtan, 2006) | |
| Coot | (Emsley et al, 2010) | |
| Phenix | (Adams et al, 2010) | |
| Prism | Graphpad | |
| Sciex OS v.1.6.1 | SCIEX | |
| BioPharmaView v.3.0.1 | SCIEX | |
| Proteome Discoverer 2.2 | ThermoFisher Scientific | |
| NITPIC | (Keller et al, 2012) | |
| SEDPHAT | (Houtman et al, 2007) | |
| GUSSI | (Brautigam, 2015) | |
| Spark Control | Tecan | |
| FIJI | (Schindelin et al, 2012) | |
| PyMol | Schrödinger | |
| WebLogo server | (Crooks et al, 2004) | |
| **Other** | | |

## Generation of plasmids

The SidBait vector (6xHis-Avitag-Ub-MCS-Ub) was synthesized as a gBlock (IDT) and cloned into a pET-28a vector. The SNAPf and Tpx2[1-43] constructs were synthesized as gBlocks (IDT) and the p38β construct was synthesized as a gene fragment (Twist Bioscience) and cloned into the SidBait vector. *E. americana* HopBF1 was PCR amplified from a pET-28a-based vector (Lopez et al, 2019) and cloned into the SidBait vector. Codon optimized SdeA (178-1100) was subcloned into ppSumo (a modified pET-28a based vector containing a 6X His tag and the yeast smt3 CDS) and used in SidBait experiments to cross-link baits to targets. Codon optimized SidE (SdeA[519-1100]) and *E. coli* BirA (amplified from BL21 gDNA) were cloned (untagged) into pETDuet-1 and used to ADP-ribosylate and biotinylate the SidBait proteins. RavB coding sequences were amplified by PCR using *Legionella pneumophila* genomic DNA (gDNA) as a template and cloned into pEGFP-C1, pProEx2, and ppSumo. Human CAMKIIδ[11-309] was PCR amplified from cDNA and cloned into ppSumo. CBR115 (Glu964–Ser1078 of Human CARMIL1) (Hernandez-Valladares et al, 2010) was synthesized as a gBlock (IDT) and cloned into ppSumo. FcγRIIA was PCR amplified from cDNA and cloned into a pcDNA vector containing a C-terminal FLAG tag.

All mutations were made using QuikChange site directed mutagenesis, and primers were designed with the online Agilent QuikChange Primer Design webpage.

## SidBait protein expression and purification

All pET-28a SidBait constructs (Ub-POI-Ub, Ub-SNAPf-Ub, or the Ub-MCS-Ub control) were co-transformed with pET-Duet1-BirA-SdeA[519-1100] into Rosetta (DE3) *E. coli*. Cells were grown in 1 L LB in the presence of 50 µg ml⁻¹ kanamycin, 100 µg ml⁻¹ ampicillin and 1 mM biotin at 37 °C and induced at an $OD_{600}$ of ~0.7 with 0.4 mM IPTG. Proteins were expressed overnight at 18 °C. Cultures were centrifuged at $3000 \times g$ for 10 min, and the bacterial pellets were resuspended in lysis buffer (50 mM Tris-HCl, pH 8.0, 300 mM NaCl, 25 mM imidazole, 1 mM DTT, 1 mM PMSF). Resuspended cells were lysed by sonication and the lysates were cleared by centrifugation at $30,000 \times g$ for 30 min. The lysate was passed over Ni-NTA beads, which were washed with lysis buffer. The protein samples were eluted with elution buffer (50 mM Tris, pH 8.0, 300 mM NaCl, 300 mM imidazole, 1 mM DTT) and further purified by size-exclusion chromatography using a Superdex HiLoad S75 16/600 column (Cytiva) into 50 mM Tris, pH 8.0, 300 mM NaCl, 1 mM DTT. Complete ADP-ribosylation of ubiquitins and biotinylation of the AviTag was ensured by intact mass spectrometry analysis. Protein samples were flash frozen and stored at −80 °C until use.

## SidBait protein laddering assay

Five µg of SidBait protein were mixed with 5 µg SidE (SdeA[178-1100]) in 20 µl reactions in TBS-DTT buffer (50 mM Tris-HCl pH 8.0, 150 mM NaCl, 1 mM DTT) for 1 h at room temperature. Reactions were terminated with 5x SDS-PAGE loading buffer (62.5 mM Tris-PO₄ pH 6.8, 50% w/v glycerol, 6.25% SDS, 0.1% bromophenol blue, 5% 2-mercaptoethanol), separated by SDS-PAGE and visualized by Coomassie staining.

## SidE protein expression and purification

6xHis-ppSumo-SdeA[178-1100] was transformed into Rosetta (DE3) *Escherichia coli*. Cells were grown in 1 L LB in the presence of 50 µg ml⁻¹ kanamycin at 37 °C and protein expression was induced at an $OD_{600}$ of ~0.7 with 0.4 mM IPTG. Protein was expressed overnight at 18 °C. Cultures were centrifuged at $3000 \times g$ for 10 min, and the bacterial pellets were resuspended in lysis buffer (50 mM Tris-HCl, pH 8.0, 300 mM NaCl, 25 mM imidazole, 1 mM DTT, 1 mM PMSF). Resuspended cells were lysed by sonication and the lysates were cleared by centrifugation at $30,000 \times g$ for 30 min. The lysate was passed over Ni-NTA beads, which were washed with lysis buffer. The Sumo-tagged SdeA[178-1100] was eluted with elution buffer (50 mM Tris-HCl, pH 8.0, 300 mM NaCl, 300 mM imidazole, 1 mM DTT) and cleaved overnight at 4 °C with Ulp Sumo protease. The cleaved SdeA[178-1100] was further purified by size-exclusion chromatography using a Superdex HiLoad S200 16/600 size exclusion column (Cytiva) into size exclusion buffer (50 mM Tris-HCl, pH 8.0, 150 mM NaCl, 1 mM TCEP). SdeA[178-1100] was concentrated to 5 mg/mL, aliquoted, flash frozen and stored at −80 °C until use.

## RavB and CAMKIIδ expression and purification

6xHis-ppSumo-RavB and 6xHis-ppSumo-CAMKIIδ[11-309] were transformed into Rosetta (DE3) *Escherichia coli*. Cells were grown in 1 L LB in the presence of 50 µg ml⁻¹ kanamycin at 37 °C and protein expression was induced at an $OD_{600}$ of ~0.7 with 0.4 mM IPTG. Proteins were expressed overnight at 18 °C. Cultures were centrifuged at $3000 \times g$ for 10 min, and the bacterial pellets were resuspended in lysis buffer (50 mM Tris-HCl, pH 8.0, 300 mM NaCl, 25 mM imidazole, 1 mM DTT, 1 mM PMSF). Resuspended cells were lysed by sonication and the lysates were cleared by centrifugation at $30,000 \times g$ for 30 min. The lysate was passed over Ni-NTA beads, which were washed with lysis buffer. The protein samples were eluted with elution buffer (50 mM Tris-HCl, pH 8.0, 300 mM NaCl, 300 mM imidazole, 1 mM DTT) and cleaved overnight at 4 °C with Ulp Sumo protease. Cleaved proteins were further purified by size-exclusion chromatography using a Superdex 75 10/300 GL size exclusion column (GE) into size exclusion buffer (50 mM Tris-HCl, pH 8.0, 150 mM NaCl, 1 mM TCEP). Protein samples were flash frozen and stored at −80 °C until use.

## CBR115 expression and purification

6xHis-ppSumo-CBR115 was transformed into Rosetta (DE3) *Escherichia coli*. Cells were grown in 1 L LB in the presence of 50 µg ml⁻¹ kanamycin at 37 °C and protein expression was induced at an $OD_{600}$ of ~0.7 with 0.4 mM IPTG. Proteins were expressed overnight at 18 °C. Cultures were centrifuged at $3000 \times g$ for 10 min, and the bacterial pellets were resuspended in lysis buffer (50 mM Tris-HCl, pH 8.0, 300 mM NaCl, 25 mM imidazole, 1 mM DTT, 1 mM PMSF). Resuspended cells were lysed by sonication and the lysates were cleared by centrifugation at $30,000 \times g$ for 30 min. The lysate was passed over Ni-NTA beads, which were washed with lysis buffer. The proteins were cleaved overnight at 4 °C with Ulp Sumo protease on the Ni-NTA beads. The supernatant was collected and were further purified by size-exclusion chromatography using a Superdex 75 10/300 GL size

exclusion column (GE) into size exclusion buffer (50 mM Tris-HCl, pH 8.0, 150 mM NaCl, 1 mM DTT). Protein samples were flash frozen and stored at −80 °C until use.

## CapZ expression and purification

Chicken CapZα/β constructs (full length α/β for binding and polymerization assays; full length a and CapZβ$^{1-244}$ for crystallography experiments) in a pETDuet vector (a gift from Michael Rosen) were transformed into Rosetta (DE3) *Escherichia coli*. Cells were grown in 1 L LB in the presence of 100 μg ml$^{-1}$ ampicillin at 37 °C and protein expression was induced at an OD$_{600}$ of ~0.7 with 0.4 mM IPTG. Proteins were expressed overnight at 18 °C. Cultures were centrifuged at 3000 × *g* for 10 min, and the bacterial pellets were resuspended in lysis buffer (50 mM Tris-HCl, pH 8.0, 300 mM NaCl, 25 mM imidazole, 1 mM DTT, 1 mM PMSF). Resuspended cells were lysed by sonication and the lysates were cleared by centrifugation at 30,000 × *g* for 30 min. The lysate was passed over Ni-NTA beads, which were washed with lysis buffer. The protein samples were eluted with elution buffer (50 mM Tris-HCl, pH 8.0, 300 mM NaCl, 300 mM imidazole, 1 mM DTT). CapZ was cleaved overnight at 4 °C with TEV protease. CapZ constructs were further purified by size-exclusion chromatography using a Superdex 200 HiLoad 16/600 size exclusion column (Cytiva) into size exclusion buffer (50 mM Tris-HCl, pH 8.0, 150 mM NaCl). Protein samples were concentrated to 10 mg/mL and flash frozen and stored at −80 °C until use.

## SidBait small molecule experiments

CLP-tagged small molecules were resuspended from powder in DMSO to a concentration of 10 mM. Preparation of SidBait-SNAPf$^{WT}$-SidBait and SidBait-SNAPf$^{C145A}$-SidBait probes were performed in parallel. Each SidBait-SNAPf-SidBait construct was diluted to 50 μM in 1x TBS, 1 mM TCEP and incubated with a 5x molar excess of the CLP-tagged small molecule at 37 °C for 30 min. Probes were desalted into 1x TBS, 1 mM TCEP to remove excess small molecule using a Zeba 7 K MWCO spin desalting column (Thermo Fisher). Full incorporation of CLP-tagged molecules into the SidBait-SNAPf$^{WT}$-SidBait probes, and lack of incorporation into the SidBait-SNAPf$^{C145A}$-SidBait control, were ensured by intact mass spectrometry.

Confluent cells in 15 cm dishes were washed twice with 10 mL of ice-cold PBS. After addition of 1.5 mL of 2x lysis buffer (100 mM Tris-HCl pH 7.5, 300 mM NaCl, 2 mM DTT, 2x Roche cOmplete protease inhibitor cocktail, 2 mM EDTA, 2% Triton X-100) per plate, cells were lysed with a cell scraper, aliquoted into 1.5 mL centrifuge tubes and cleared by centrifugation at 21,300 × *g* for 20 min. The supernatant was diluted to 5 mg/mL in 1x lysis buffer and used for subsequent experiments.

Cell lysates (300 μL) were added to SidBait probes such that the final probe concentration was 2–5 μM. Following incubation for 30–60 min, SdeA (1 μL from a 5 mg/mL stock) was added to the samples to initiate the cross-linking reaction and samples were incubated on ice for a further 2–12 h. Streptavidin-agarose beads (50 mL of 50% slurry) were added to the reactions and samples were nutated overnight at 4 °C. Beads were then spun at 800 g for 1 min at 4 °C and washed successively with wash buffer 1 (2% SDS), wash buffer 2 (50 mM HEPES pH 7.5, 0.1% sodium deoxycholate,

1% Triton X-100, 1 mM EDTA, 500 mM NaCl), wash buffer 3 (10 mM Tris-HCl pH 8.0, 0.5% sodium deoxycholate, 0.5% NP-40, 1 mM EDTA, 250 mM LiCl) and wash buffer 4 (50 mM Tris-HCl pH 7.5) before mass spectrometry analysis.

## SidBait protein experiments

HEK293A lysates were prepared as described above. Cell lysates (300 μL) were added to SidBait proteins (30 μL of 30 μM protein) on ice. Following incubation for 30–60 min, SdeA (1 μL from a 5 mg/mL stock) was added to the samples to initiate the cross-linking reaction and samples were incubated on ice for an additional 2–12 h. Samples were bound to streptavidin-agarose beads and washed as described above before mass spectrometry analysis.

## SidBait competition assay

Lysates from HEK293A cells and SidBait probes were prepared as described above. Before addition of the probe to lysate, increasing concentrations of free-probe (native drug or untagged-protein) were added to the 300 μL aliquots of the lysate and mixed well by pipetting. The SidBait probe was then added to the lysates, incubated, and bound to beads as described above. The following day, before the first bead wash, the unbound lysate was collected. 40 μg of total unbound fraction was combined with 5x SDS-PAGE loading buffer, resolved by SDS-PAGE, transferred to a nitrocellulose membrane and immunoblotted with the indicated antibodies.

## Animal tissue experiments

Mouse hearts and kidneys were washed in ice-cold PBS, dounced in 2x SidBait lysis buffer and further lysed by sonication. The lysates were cleared by centrifugation at 21,300 × *g* for 20 min. Small molecule SidBait experiments in animal tissue lysates were performed as above.

## Yeast experiments

BY4741 strain *Saccharomyces cerevisiae* was grown in YDP media at 30 °C until confluency. Cells were pelleted at 3000 × *g* for 10 min and resuspended in an equivalent volume of ice-cold SidBait lysis buffer (as detailed above). Yeast were lysed by vortexing using glass bead beating in flat bottom microcentrifuge tubes. Cleared cell lysates were obtained by two spins at 3000 × *g* (2 min at 4 °C) and 20,000 × *g* (10 min at 4 °C). Lysates were then equilibrated to 5 mg/ml for subsequent SidBait experiments.

## *Xenopus* embryo experiments

Adult *Xenopus* were housed in X-Rack recirculating System (Aquatic Habitats) following standard recommendations. The water was monitored daily for temperature, salt, pH, nitrate and nitrite. All animal procedures were approved by the Institutional Animal Use and Care (IACUC) committee at the University of Massachusetts Amherst (Protocol #5609).

Fertilized *Xenopus laevis* embryos were obtained by in vitro fertilization (Sive and Harland, 2023) and were injected with 5 nL of bait (10 μM) followed immediately by 5 nL of SdeA$^{178-1100}$ (5 mg/

mL) at the one-cell stage and incubated for one hour at 14 °C. Two sets of embryos from different females were injected to obtain two biological replicates. Proteins were extracted in 1x MBS (Modified Barth Solution; 88 mM NaCl, 1 mM KCl, 2.4 mM NaHCO$_3$, 15 mM HEPES pH 7.6, 0.3 mM CaNO$_3$·4H$_2$O, 0.41 mM CaCl$_2$·6H$_2$O, 0.82 mM MgSO$_4$) containing 1% Triton X-100, Halt Protease phosphatase inhibitor (Thermo Fisher) and 5 mM EDTA. Yolk and debris were spun down at 16,000 × $g$ at 4 °C for 30 min. The soluble fraction was incubated with 10 μl of streptavidin-magnetic beads (Pierce) for 1 h at RT. Beads were washed as above prior to mass spectrometry analysis.

## Intact mass spectrometry analysis

Protein samples were analyzed by LC–MS, using a Sciex X500B Q-TOF mass spectrometer coupled to an Agilent 1290 Infinity II HPLC. Samples were injected onto a POROS R1 reverse-phase column (2.1 mm × 30 mm, 20 μm particle size, 4000 Å pore size) and desalted. The mobile phase flow rate was 300 μl min$^{-1}$ and the gradient was as follows: 0–3 min, 0% B; 3–4 min, 0–15% B; 4–16 min, 15–55% B; 16–16.1 min, 55–80% B; 16.1–18 min, 80% B. The column was then re-equilibrated at the initial conditions before the subsequent injection. Buffer A contained 0.1% formic acid in water and buffer B contained 0.1% formic acid in acetonitrile.

The mass spectrometer was controlled by Sciex OS v.1.6.1 using the following settings: ion source gas 1, 30 psi; ion source gas 2, 30 psi; curtain gas, 35; CAD gas, 7; temperature, 300 °C; spray voltage, 5500 V; declustering potential, 80 V; collision energy, 10 V. Data were acquired from 400–2000 Da with a 0.5 s accumulation time and 4 time bins summed. The acquired mass spectra for the proteins of interest were deconvoluted using BioPharmaView v.3.0.1 (Sciex) to obtain the molecular mass values. The peak threshold was set to ≥5%, reconstruction processing was set to 20 iterations with a signal-to-noise threshold of ≥20 and a resolution of 2500.

## On-bead digestion of SidBait samples and mass spectrometry analysis

Proteins on-beads were washed thoroughly prior to enzymatic digestion and LC-MS/MS analysis. Reduction and alkylation of cysteines was carried out first with 10 mM DTT (1 h, 56 °C) and 50 mM iodoacetamine (45 min, RT in the dark), respectively. Proteins on-beads were digested overnight at 37 °C with sequencing grade trypsin in 50 mM ammonium bicarbonate. The next day, tryptic peptides were acidified with 5% trifluoroacetic acid to stop digestion and desalted via solid phase extraction (SPE). LC-MS/MS experiments were performed on a Thermo Scientific EASY-nLC liquid chromatography system coupled to a Thermo Scientific Orbitrap Fusion Lumos mass spectrometer. To generate MS/MS spectra, MS1 spectra were first acquired in the Orbitrap mass analyzer (resolution 120,000). Peptide precursor ions were then isolated and fragmented using high-energy collision-induced dissociation (HCD). The resulting MS/MS fragmentation spectra were acquired in the ion trap. Label-free quantitative searches were performed using Proteome Discoverer 2.2 software (Thermo Scientific). Samples were searched against all entries included in the Human Uniprot database. Modifications included carbamidomethylation of cysteine (+57.021 Da), oxidation of methionine

(+15.995 Da), and acetylation of peptide N-termini (+42.011 Da). Precursor and product ion mass tolerances were set to 10 ppm and 0.6 Da, respectively. Peptide spectral matches were adjusted to a 1% false discovery rate (FDR) and additionally proteins were filtered to a 5% FDR.

Protein abundances were quantified in SidBait-protein vs SidBait-empty samples by comparing area values of precursor ions. Abundance values were normalized across samples based on the total peptide amount identified in each. Samples with missing values were assigned a value equal to the average of the lower 1% of all abundance values. Enrichment scores were calculated as a ratio of the SidBait-protein:SidBait-empty abundances. Gene names and enrichment scores were plotted using Prism software.

## Crystallization, data collection and structure determination

CAMKIIδ$^{kd}$ was prepared from *E. coli* as described above and diluted to 10 mg/mL in 10 mM Tris-HCl pH 8.0, 150 mM NaCl and 1 mM TCEP. CAMKIIδ$^{kd}$:ribociclib was prepared by incubation of 350 μM CAMKIIδ$^{kd}$ with 800 μM ribociclib for 1 h. CAMKIIδ$^{kd}$:ribociclib crystals were grown by the sitting drop vapor diffusion method overnight at 4 °C in 24-well Cryschem trays using a 1:1 ratio of protein/reservoir solution containing 0.2 M CaCl$_2$, 20% w/v PEG 3350. Growth of single crystals was initiated the following day by micro-seeding a Cryschem tray containing a 1:1 ratio of protein/reservoir solution containing 0.2 M CaCl$_2$, 10% w/v PEG 3350. Wells were allowed to equilibrate for 24–48 h and crystal growth was initiated by micro-seeding. CAMKIIδ$^{kd}$:ribociclib crystals were cryo-protected with 0.2 M CaCl$_2$, 0.15 M NaCl, 12% (w/v) PEG 3350, 0.7 mM ribociclib and 35% (w/v) ethylene glycol, diffracted to a minimum Bragg spacing ($d_{min}$) of 2.35 Å and exhibited the symmetry of space group P2$_1$2$_1$2$_1$ with cell dimensions of a = 46.3 Å, b = 82.8 Å, c = 172.9 Å, and contained two CAMKIIδ$^{kd}$:ribociclib per asymmetric unit. RavB$^{108-C}$:CapZα:CapZβ$^{1-244}$ was prepared by expression and purification from *E. coli* as described above and diluted to 10 mg/mL in 10 mM Tris-HCl pH 8.0, 150 mM NaCl and 1 mM TCEP. RavB:CapZ crystals were grown by the sitting drop vapor diffusion method at 20 °C in 24-well Cryschem trays using a 1:1 ratio of protein/reservoir solution containing 0.2 M ammonium formate, 19% w/v PEG 3350, 20 mM 1,2-dioctanoyl-sn-glycero-3-(phosphoinositol-3-phosphate). Wells were allowed to equilibrate for 24–48 h and crystal growth was initiated by micro-seeding. RavB:CapZ crystals were cryo-protected with 10 mM Tris pH 8.0, 0.2 M ammonium formate, 0.15 M NaCl, 21% w/v PEG 3350 and 30% (w/v) ethylene glycol, diffracted to a minimum Bragg spacing ($d_{min}$) of 2.00 Å and exhibited the symmetry of space group P2$_1$ with cell dimensions of a = 66.5 Å, b = 55.3 Å, c = 77.9 Å, β = 107.1° and contained one RavB:CapZ per asymmetric unit. Diffraction data for CAMKIIδ$^{kd}$:ribociclib were collected at beamline BL12-2 at the Stanford Synchrotron Radiation Lightsource (SLAC National Accelerator Laboratory, Menlo Park, California, USA). Diffraction data for RavB:CapZ crystals were collected on a Rigaku MicroMax-003 instrument outfitted with a copper sealed tube and a HyPix-6000HE direct photon detector. All diffraction data were processed in the program HKL-3000 (Minor et al, 2006) with applied corrections for effects resulting from absorption in a crystal and for radiation damage (Borek et al, 2003; Otwinowski et al, 2003), the

calculation of an optimal error model, and corrections to compensate the phasing signal for a radiation-induced increase of non-isomorphism within the crystal (Borek et al, 2010; Borek et al, 2013). Phases for CAMKIIδ$^{kd}$:ribociclib were calculated via molecular replacement using residues 10–309 of PDB ID 3SOA (Chao et al, 2011) as a search model. Phases for RavB:CapZ were calculated via molecular replacement using PDB ID 3AA7 (Takeda et al, 2010) as a search model for the CapZα:CapZβ dimer. Molecular replacement for both structures was performed using the program PHASER (McCoy et al, 2007). A model for the RavB residues was automatically generated in the program Buccaneer (Cowtan, 2006). Completion of models for both structures was performed by multiple cycles of manual rebuilding in the program Coot (Emsley et al, 2010) and refinement in the program Phenix (Adams et al, 2010). Positional and isotropic atomic displacement parameter (ADP) as well as TLS ADP refinement was performed in the program Phenix with a random 7.0% of all dataset aside for an $R_{free}$ calculation. Data collection and structure refinement statistics are summarized in Appendix Table S1.

## Bacterial strains, cell lines, and culture media

*L. pneumophila* strains were grown in ACES-buffered yeast extract (AYE) broth or on ACES-buffered charcoal yeast extract (CYE) agar plates as previously described (Chatfield and Cianciotto, 2013; Hsieh et al, 2021). *E. coli* strains were grown in Luria-Bertani (LB) broth or on LB agar plates supplemented with 100 µg/mL ampicillin or 50 µg/mL kanamycin when appropriate.

*L. pneumophila* Philadelphia-1 wild-type (WT) strain Lp02 was a gift from Ralph Isberg. Lp02-mcherry was generated by expressing mCherry CDS cloned into pJB908 (Sexton et al, 2005), a gift from Ralph Isberg.

COS-7, HeLa, and HEK293 cells were cultured in DMEM/High glucose with L-glutamine (ThermoFisher) supplemented with 10% FBS, 1% penicillin-streptomycin and incubated at 37 °C with 5% $CO_2$.

H9 cells were cultured in mTeSR Plus according to standard protocols and were seeded in 6-well plates to start differentiation. Upon reaching 80–90% confluency, media was replaced with 3 µM CHIR99021 supplemented CDM3 (RPMI 1640; 0.5 mg/ml human albumin, ScienCell OsrHSA; 211 µg/ml L-ascorbic acid 2-phosphate) for 48 h. Subsequently, media was changed to 2 µM Wnt-C59 supplemented CDM3 for 48 h. Then, media was changed every 2 days with CDM3 alone. On day 8 post-differentiation, samples were changed to RPMI 1640 + B27. Cells were analyzed for beating between Days 12–14 post differentiation. For SidBait experiments using cardiomyocytes, differentiations were performed as described with the following modification: on day 10 post-differentiation, 1 million cells were passaged onto 10 cm dishes coated in Matrigel and incubated with RPMI 1640 + B27 + 2 µM CHIR99021 to allow for expansion of cardiomyocytes.

Cell lines were neither authenticated nor tested for mycoplasma contamination.

## In vitro kinase activity assays

In vitro kinase assays using ribociclib were performed by Reaction Biology. IC$_{50}$ values for ribociclib were measured against CaMKIIδ and CDK4/cyclinD1with a γ-$^{33}$P radioactivity assay. Human

CaMKIIδ was incubated with $Ca^{2+}$-Calmodulin (1 µM), a synthetic peptide substrate (KKLNRTLSFAEPG, 20 µM) and varying concentrations of ribociclib for 20 min at room temperature in base reaction buffer (20 mM HEPES pH 7.5, 10 mM $MgCl_2$, 1 mM EGTA, 0.01% Brij35, 0.02 mg/ml BSA, 0.1 mM $Na_3VO_4$, 2 mM DTT, 1% DMSO). Human CDK4/cyclinD1 was incubated with Rb protein (3 µM) for 20 min at room temperature in base reaction buffer. Kinase reactions were initiated with 10 µM $^{33}$P-ATP and run for 2 h. Phosphorylation of substrates was measured after blotting on P81 filter membranes.

## Ribociclib inhibition of human cardiomyocytes

Cardiomyocytes were differentiated and passaged as described above until 80% confluency and then treated with DMSO or indicated concentrations of ribociclib (Selleckchem/Adooq) for 1 h and then stimulated with DMSO, 1 mM $Ca^{2+}$, or 1 µM isoproterenol (Sigma-Aldrich) to induce CaMKII activity for 30 min. Cells were lysed directly on the plate with 2x SDS-PAGE loading buffer, boiled, resolved by SDS-PAGE, transferred to a nitrocellulose membrane, and immunoblotted with the indicated antibodies.

## Heartbeat analysis

Cardiomyocytes were differentiated and passaged as described until 80% confluency and then stimulated with an additional 1 mM $Ca^{2+}$ to induce CaMKII activity for 30 min and then treated with 1 µM ribociclib (Selleckchem/Adooq), 1 µM KN93 (Selleckchem), or 1 µM flavipiridol (Selleckchem) for one hour. Cells were visualized using a Zeiss Primovert Microscope and 4 × 30 s videos of each well were taken with an iPhone 11 rear-facing camera mounted directly to the right eyepiece.

## Isothermal titration calorimetry

Full-length CapZ and RavB were purified as described above. For RavB$^{CPI}$ measurements, a synthetic peptide consisting of RavB residues 108–148 (Genscript) was used. Protein and peptide samples were buffer exchanged into 50 mM Tris-HCl pH 8.0, 150 mM NaCl, 1 mM TCEP. All ITC experiments were performed on a Malvern MicroCal PEAQ ITC instrument. RavB$^{WT}$ (100 µM), RavB$^{3R3A}$ (100 µM) or RavB$^{CPI}$ (300 µM) were injected into a cell containing CapZ (10 µM, 10 µM, or 30 µM, respectively) at 20 °C. The first injection was at a volume of 0.5 µL and followed by 20 injections of 1.9 µL. Data integration, fitting and error analysis were performed using NITPIC and SEDPHAT software. Results are reported as best fit with a 68.3% confidence interval. Figures were prepared using GUSSI software.

## Immunoprecipitation experiments

HEK293A cells were plated into a 6-well dish at 50% confluency. The following day, individual wells were transfected with pEGFP-C1, pEGFP-RavB$^{WT}$, or pEGFP-RavB$^{3R3A}$ using PolyJet (SignaGen SL100688) transfection reagent. Briefly, 1 µg plasmid DNA was added to 100 µL serum/antibiotic-free DMEM media and 3 µL PolyJet mixture, incubated at room temperature for 15 min, and added dropwise to the wells. The medium was replaced 5 h after transfection (DMEM containing 10% FBS, 1% penicillin-

streptomycin). After 24 h, cells were washed twice with ice-cold PBS and lysed with 500 μL mammalian lysis buffer (50 mM Tris-HCl pH 8.0, 150 mM NaCl, 1 mM DTT, 1% Triton X-100, 1 mM ETDA, 1x cOmplete Protease Inhibitor Cocktail tablet). Lysates were cleared by centrifugation (4 °C, 10 min, 21,300 × g), and 50 μL was saved for input. GFP-Catcher beads (Antibodies-online ABIN5311508) were washed twice in mammalian lysis buffer (4 °C, 2 min, 800 × g), and 20 μL packed beads per sample were added directly to the lysates and nutated overnight at 4 °C. The following day, beads were washed twice with mammalian lysis buffer (4 °C, 2 min, 800 × g), and conjugated proteins were eluted directly off the beads by boiling in 2x SDS-PAGE loading buffer for 10 min. 40 μg of input and a fourth of total immunoprecipitated material were resolved by SDS-PAGE, transferred to a nitrocellulose membrane and immunoblotted with the indicated antibodies.

## Fluorescence microscopy

Fluorescence microscopy was performed as previously described (Hsieh et al, 2021). Spinning disk confocal microscopy was performed with an Olympus IXplore SpinSR10 system (Olympus, Waltham, MA). Images were taken with a U Apo N TIRF 100x/1.49 oil objective (Olympus, Waltham, MA) and an Andor iXon Ultra 888 electron multiplier charge-coupled device (EM-CCD) camera (Oxford Instruments, Belfast, UK) when using the Olympus microscope.

For fluorescence imaging experiments, HeLa cells were seeded on an 8-well chamber cover glass at a density of $\sim 2 \times 10^4$ cells/well 1 day before transfection. The following day, individual wells were transfected with 35 ng of pEGFP-RavB$^{WT}$, pEGFP-RavB3$^{R3A}$, pEGFP-RavB$^{CTD}$, or pEGFP-RavB$^{H214A}$ and 100 ng of mTagBFP-Lamp1 (mTagBFP-Lysosomes-20 was a gift from Michael Davidson; Addgene plasmid #55263; RRID:Addgene_55263) (Rizzo et al, 2009) using TransIT-X2 (mirusbio MIR 6000) transfection reagent per manufacturer's instructions. Cells were fixed 18 h after transfection.

For imaging *L. pneumophila* infection in mammalian cells, COS-7 cells expressing IgG FC receptor FcγRIIA was used as the model. COS-7 cells seeded on collagen-coated 8-well chamber coverglass at a density of $\sim 2 \times 10^4$ cells/well 1 day before transfection. The following day, individual wells were transfected with 35 ng of pEGFP-RavB$^{H214A}$ or pEGFP-RavB3$^{R3A,H214A}$ and 100 ng of pCCF-FcγRIIA using TransIT-X2 transfection reagent per manufacturer's instructions. At the same time as transfection, *L. pneumophila*-mCherry strains were grown in AYE media. 16 h post-transfection, stationary phase *L. pneumophila*-mCherry was opsonized with anti-*L. pneumophila* antibody (ThermoFisher AWB4CE4) in opti-MEM at 37 °C for 30 min, at the ratio of $10^6$ bacteria: 70 ng antibody: 50 μL of opti-MEM. Transfected COS-7 cells were washed once with PBS and infected with opsonized stationary phase *L. pneumophila*-mCherry at an MOI of 40. Infection was synchronized by centrifugation (RT, 5 min, 500 × g), incubated for another hour in normal culture conditions (37 °C, 5% CO$_2$), then fixed.

Cells were washed once in PBS, then fixed in 4% paraformaldehyde (Electron Microscopy Sciences) diluted in PBS for 15 min at room temperature (RT), washed with PBS, permeabilized with 0.1% Triton X-100 in PBS for 15 min at RT, and blocked with 5% BSA in PBS for 10 min at RT. Indicated primary antibodies were incubated

in 5% BSA in PBS at 1:400 dilution overnight at 4 °C. The following day, primary antibody was washed three times with PBS, followed by incubation with indicated Alexa fluorochrome conjugated secondary antibodies at 1:400 dilution in 5% BSA in PBS. Samples were washed three times in PBS and visualized by spinning disk confocal microscopy. All incubations were protected from light and agitated gently.

Colocalization statistics were computed by using Fiji/ImageJ built in Coloc 2 function. Regions of interest (ROI) were manually drawn using the polygon selections tool around the entirety of a single cell (excluding nucleus for RavB$^{H214A}$ images). Coloc 2 (Single cell ROI, Threshold regression = Bisection, PSF = 3.0, Costes randomizations = 100) was run between the specified color channels, and the Pearson's R value (above threshold) was noted and plotted in PRISM.

## Generation of rabbit polyclonal RavB antibody

His-tagged RavB was purified from Rosetta (DE3) *E. coli* by Ni-NTA affinity purification followed by size-exclusion chromatography. Proteins were sent to Cocalico Biologicals, Inc. for inoculation of rabbits. Anti-serum was received and the anti-RavB antibodies were partially purified by ammonium sulfate precipitation (Kent, 1999). Briefly, 15 mL of the anti-serum was centrifuged (4 °C, 30 min, 10,000 × g), and incubated with 35 mL saturated ammonium sulfate (0.5 L 0.2 M sodium borate pH 8, 160 mM NaCl + 400 g (NH$_4$)2SO$_4$, heated until dissolved, cooled at 4 °C overnight), with mixing for 2 h at 4 °C. The mix was then spun (4 °C, 30 min, 10,000 × g) and supernatant discarded. The pellet was dissolved in 15 mL of BBS (0.2 M sodium borate pH 8, 160 mM NaCl) spun (4 °C, 10 min, 10,000 × g) and the supernatant collected.

RavB was purified as described above but using 50 mM HEPES instead of 50 mM Tris-HCl as buffer to omit primary amines. After size-exclusion chromatography, the protein was buffer exchanged into 0.2 M NaHCO$_3$ pH 8.3, 0.5 M NaCl. A HiTrap NHS-activated High Performance column (Cytiva 17071701) was washed with 6 mL of 1 mM HCl, then RavB was injected and pumped back and forth for 30 min. The column was then injected with 1 mL of 10 mM Tris pH 7.5 and deactivated with 6 mL Buffer A (0.5 M ethanolamine pH 8.3, 0.5 M NaCl) then 6 mL Buffer B (0.1 M acetic acid pH 4, 0.5 M NaCl), repeated three times for six total washes.

The protein-conjugated column was then washed with 10 mL of 10 mM Tris-HCl pH 7.5, then 10 mL of 100 mM glycine pH 2.5, 10 mM BBS. Fractionated serum was then loaded, and subsequently washed with 20 mL BBS, then 20 mL 10 mM Tris-HCl pH 7.5, 0.5 M NaCl. Finally, the antibody was eluted with 9 mL of 100 mM glycine pH 2.5 directly into a tube containing 1 M Tris 8.0. The antibody was concentrated using a centrifugal filter, sodium azide was added to final concentration of 0.02%, and the anti-RavB antibodies were stored at −20 °C until use.

## PIP strip lipid binding assay

PIP-strip (Echelon Biosciences) binding assay was conducted according to manufacturer's instructions. All steps are performed with agitation at room temperature unless specified otherwise. Briefly, the membrane was blocked with 3% BSA in TBST (50 mM Tris-HCl pH 8.0, 150 mM NaCl, 0.1% Tween-20) for 1 h. Purified RavB was diluted in 3% BSA to 1 μg/mL, and 5 mL was added to

blocked membrane for 1 h. The membrane was washed with TBST 3 times for 5 min each. Rabbit anti-RavB antibody was added at 1:1000 dilution in 3% TBST, and incubated overnight at 4 °C. On the following day, the membrane was washed 3 times with TBST, and secondary anti-rabbit HRP antibody at 1:2000 dilution in 3% TBST was added for 1 h. This membrane was then washed 3 times with TBST, incubated with SuperSignal West Pico PLUS chemiluminescent substrate for 1 min, dried, and exposed on autoradiography film.

### Liposome sedimentation assay

Liposomes containing POPC as carrier and various phospholipids were generated by mixing chloroform stocks of POPC and lipid (POPC, PI, PI3P, PI4P, PI5P, PI34P$_2$, PI35P$_2$, PI45P$_2$, PIP$_3$) (Echelon Biosciences) to form final concentration of 12.75 mM and 2.25 mM, respectively, in 1 mL volume. The lipids were then dried using a vacufuge, resuspended in 1 mL H$_2$O, and vortexed for 5 min. Liposomes were then subject to 5 freeze-thaw cycles with liquid nitrogen. Purified RavB protein was centrifuged (4 °C, 15 min, 21,300 × g) to exclude any aggregates. 2 μg protein was incubated with 112 μM lipid (1:20 dilution of liposome stock), in 20 μL of 50 mM Tris-HCl pH 8.0, 150 mM NaCl, 1 mM DTT for 1 h at room temperature. Liposomes were sedimented by centrifugation (4 °C, 30 min, 21,300 × g). The supernatant was combined with 5x SDS-PAGE loading buffer, while the pellet was solubilized with 20 μL of 1x SDS-PAGE loading buffer. The samples were resolved by SDS-PAGE and stained with Coomassie to determine the presence of protein.

### Actin decapping and polymerization assays

Monomeric non-muscle actin (Cytoskeleton APHL99) was prepared following manufacturer's instructions. Briefly, the desiccated protein was reconstituted by adding deionized water and pipetting on ice for 10 min. Actin was then diluted to 0.4 mg/ml in general actin buffer supplemented with ATP (5 mM Tris-HCl pH 8.0, 0.2 mM CaCl$_2$, 0.2 mM ATP), and incubated on ice for 1 h to depolymerize any oligomers. Any remaining oligomers were excluded by centrifugation (4 °C, 15 min, 21,300 × g), and the supernatant containing G-actin was collected.

Generation of short, uni-sized actin filaments decorated with capping protein (Funk et al, 2021) was adapted from previous methods. Briefly, 500 μg actin monomers were incubated with molar excess (1000 μg) purified chicken CapZ complex. After 2 min at 4 °C, 10x ME buffer was added to yield 0.5 mM MgCl$_2$, 0.2 mM EDTA final concentration for 1 min at 4 °C. Polymerization was initiated by addition of 10x KMEI buffer to yield 50 mM KCl, 1.5 mM MgCl$_2$, 1 mM EGTA, 10 mM imidazole pH 7.0 final concentration for 2 min at room temperature. The protein solution was concentrated using a centrifugal filter until 500 μL final volume where it was separated by gel-filtration over a Superdex 200 increase 10/300 GL column in 1x KMEI buffer. 20 μL from each 500 μL fraction was collected, combined with 5x SDS-PAGE loading buffer, resolved by SDS-PAGE, and stained with Coomassie to determine protein-complex location.

Fractions containing actin-CapZ complex were combined, concentrated such that final Actin-CapZ-RavB solution was under 500 μL, and incubated with either molar excess (1000 μg) purified RavB$^{WT}$ or RavB$^{3R3A}$ for 30 min at 4 °C. The protein solution was then separated by gel filtration over a Superdex 200 increase 10/300 GL column in 1x KMEI buffer. 20 μL from each 500 μL fraction was collected, combined with 5x SDS-PAGE loading buffer, resolved by SDS-PAGE, and stained with Coomassie to determine protein-complex location.

For the actin polymerization assays, 2 μM G-actin-pyrene (prepared as by manufacturer's instructions; Cytoskeleton BK003) was mixed with 50 nM CapZ and 1.25 μM of CBR115 or RavB$^{115-139}$. For controls, an equal volume of buffer was added in place of CapZ or decapping peptides. Polymerization reactions were initiated by addition of 10x KMEI buffer (final volume: 60 μL) in a flat black 384-well plate (Greiner). After a 5 s shake, polymerization reactions were measured using a Tecan Spark Cyto plate reader (ex: 360 nm; em: 420 nm; 10 nm bandwidth) in 15 s intervals.

## Data availability

Correspondence and material requests should be addressed to VST. Atomic coordinates have been deposited to the Protein Data Bank with accession codes 9BLH and 9BLI.

The source data of this paper are collected in the following database record: biostudies:S-SCDT-10_1038-S44318-025-00665-0.

## Peer review information

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

## Acknowledgements

We thank members of the Tagliabracci laboratory and Florentine Rutaganira for discussions, John Scott for the initial aliquot of CLP-alisertib probe, Ralph Isberg for Legionella strains, Chad Brautigam and Shih-Chia Tso (UTSW

Macromolecular Biophysics Resource Core Facility) for help with ITC experiments, Andrew Lemoff (UTSW Proteomics Core Facility) for help with intact mass spectrometry, Helen Aronovich (UTSW Structural Biology Laboratory) for help with crystallography screening and Oguz Can Koc and Matthew J. Dunn for help with CaMKII assays. Use of the Stanford Synchrotron Radiation Lightsource, SLAC National Accelerator Laboratory, is supported by the U.S. Department of Energy, Office of Science, Office of Basic Energy Sciences under Contract No. DE-AC02-76SF00515. The SSRL Structural Molecular Biology Program is supported by the DOE Office of Biological and Environmental Research, and by the National Institutes of Health, National Institute of General Medical Sciences (P30GM133894). The contents of this publication are solely the responsibility of the authors and do not necessarily represent the official views of NIGMS or NIH. This work was funded by NIH Grants DP2GM137419 (VST), K99GM147532 (TH), Welch Foundation Grants I-1911 (VST) and I-1704 (NMA), and the Howard Hughes Medical Institute (HHMI, VST). VST is a Michael L. Rosenberg Scholar in Medical Research, a CPRIT Scholar (RR150033), a Searle Scholar and an investigator of the HHMI.

## Author contributions

**James S Ye**: Conceptualization; Data curation; Formal analysis; Investigation; Visualization; Methodology; Writing—original draft; Writing—review and editing. **Abir Majumdar**: Conceptualization; Data curation; Formal analysis; Validation; Investigation; Methodology; Writing—original draft; Writing—review and editing. **Brenden C Park**: Conceptualization; Data curation; Formal analysis; Investigation; Methodology; Writing—review and editing. **Miles H Black**: Conceptualization; Data curation; Formal analysis; Investigation; Methodology; Writing—review and editing. **Ting-Sung Hsieh**: Conceptualization; Data curation; Formal analysis; Investigation; Visualization; Writing—review and editing. **Adam Osinski**: Data curation; Formal analysis; Investigation; Writing—review and editing. **Kelly A Servage**: Data curation; Investigation; Methodology; Writing—review and editing. **Kartik Kulkarni**: Resources; Investigation. **Jacinth Naidoo**: Resources; Investigation. **Neal M Alto**: Supervision. **Margaret M Stratton**: Resources; Validation; Writing—review and editing. **Dominique Alfandari**: Resources; Investigation; Methodology. **Joseph M Ready**: Resources. **Krzysztof Pawłowski**: Conceptualization; Data curation; Software; Formal analysis; Methodology; Writing—original draft; Writing—review and editing. **Diana R Tomchick**: Resources; Investigation. **Vincent S Tagliabracci**: Conceptualization; Supervision; Funding acquisition; Investigation; Methodology; Writing—original draft; Project administration; Writing—review and editing.

Source data underlying figure panels in this paper may have individual authorship assigned. Where available, figure panel/source data authorship is listed in the following database record: biostudies:S-SCDT-10_1038-S44318-025-00665-0.

## Disclosure and competing interests statement

VST, MHB, and BCP are listed as inventors on US patent number 17684639 on the SidBait technology. The remaining authors declare no competing interests.

# Expanded View Figures

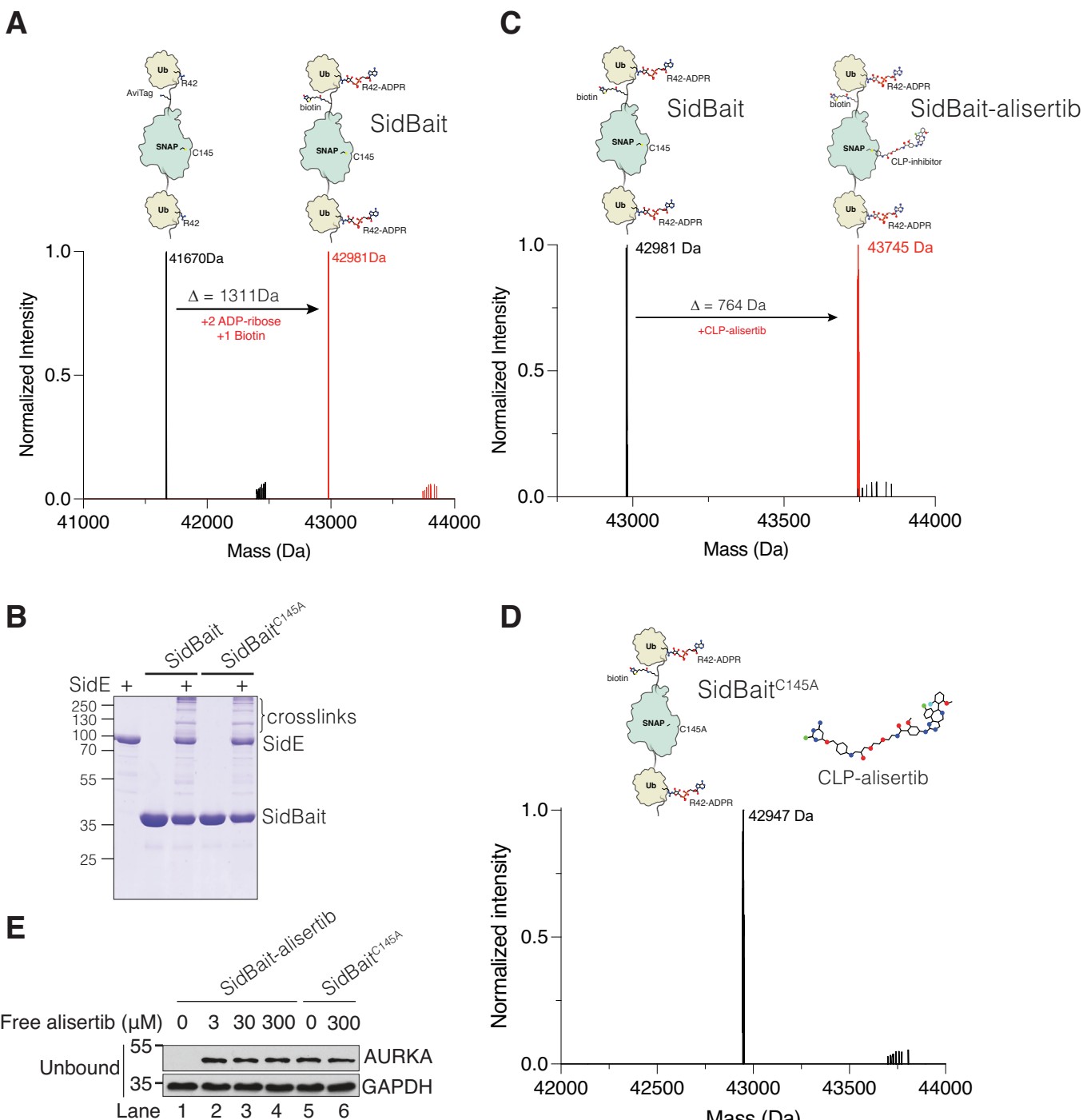

**Figure EV1. SidBait identifies targets of small molecules.**

(A) Intact mass spectrum of unmodified SidBait (left, black) and SidBait which contains a biotin and two ADP-ribose molecules (right, red). (B) NAD⁺-independent SidE autoubiquitination of the SidBait probe. The SidBait probe was incubated with SidE^PDE and the reaction products were separated by SDS-PAGE and visualized by Coomassie staining. Cross-linking of the SidBait^C145A control (right), which cannot conjugate CLP-derivatives of small molecules, demonstrates an otherwise functional SidBait construct. (C) Intact mass spectrum of SidBait (left, black) and SidBait-alisertib (right, red). (D) Protein immunoblot of the unbound fractions following avidin enrichment of the SidBait-alisertib probe from HEK293 cell lysates that have been incubated with and without free alisertib. AURKA and GAPDH are shown. All cellular AURKA is bound to the SidBait-alisertib probe in the absence of free alisertib (lane 1). Following the addition of free alisertib, the SidBait-alisertib probe is competed off AURKA (lanes 2–4). (E) Intact mass spectrum of the SidBait^C145A control after incubation with CLP-alisertib, showing that the mutant protein cannot incorporate the small molecule. Source data are available online for this figure.

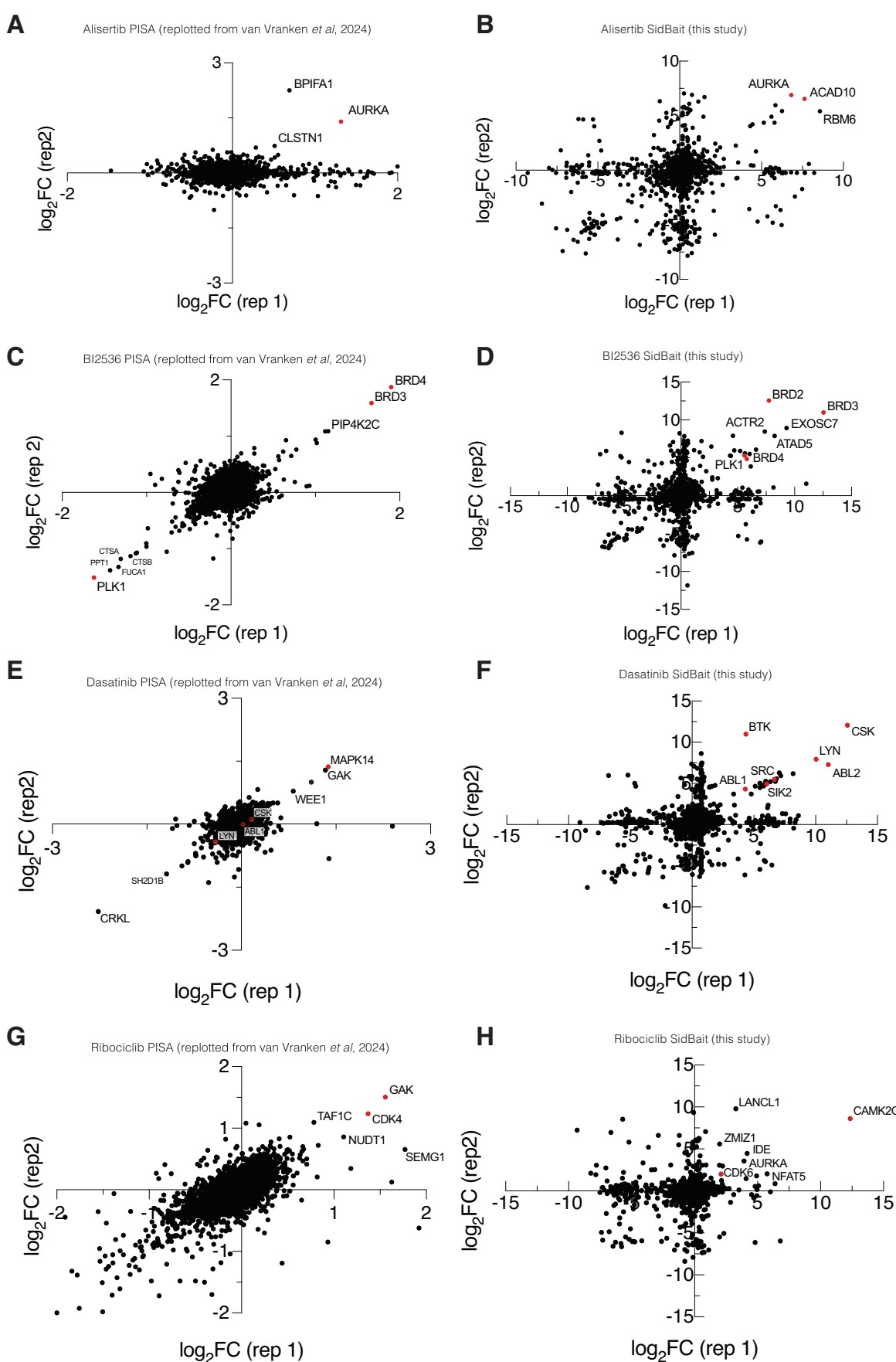

◀ **Figure EV2.** SidBait identifies small molecule targets in K562 cell lysates.

Comparison of protein targets enriched in PISA experiments (left; van Vranken et al, 2024) with SidBait experiments (right) for alisertib (**A**, **B**); BI2536 (**C**, **D**); dasatinib (**E**, **F**); and ribociclib (**G**, **H**). Data for each SidBait panel is from two independent experiments.

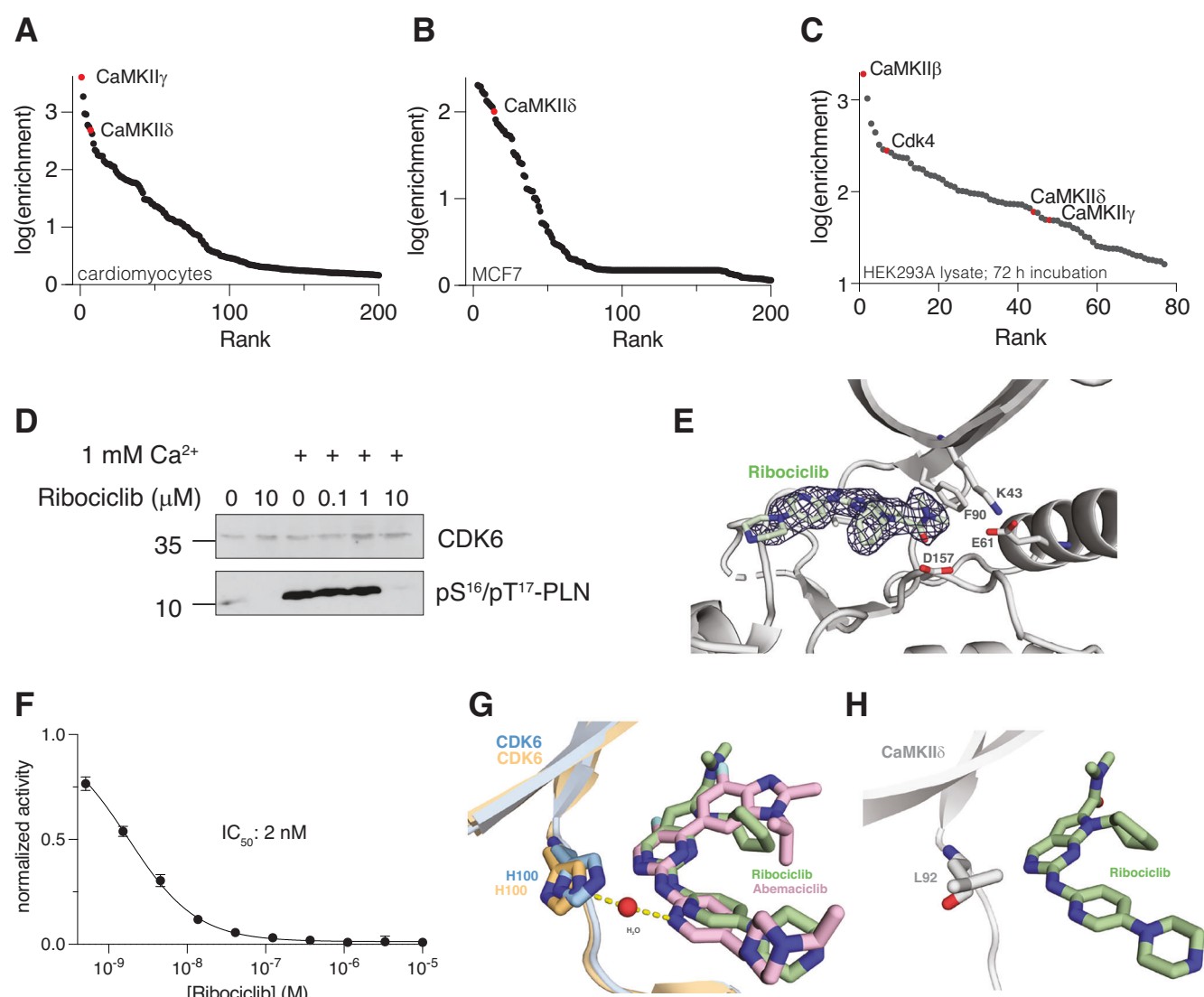

**Figure EV3. SidBait identifies CaMKII as a target of the CDK4/6 inhibitor ribociclib.**

(A, B) Plot of the fold enrichment of proteins from SidBait-ribociclib experiments in cultured cardiomyocytes (A) and MCF7 cells (B). (C) Plot of the fold enrichment of proteins from SidBait-ribociclib experiments in HEK293A cells following a 72-hour incubation with the bait. (D) Protein immunoblot of human cardiomyocyte lysates that have been stimulated with $Ca^{2+}$ in the presence of varying concentrations of ribociclib. Total CDK6 is shown as a control; $pThr^{16}/pSer^{17}$-phospholamban is shown as an indication of endogenous CaMKII activity. (E) A view of ribociclib in the active site of the CaMKII kinase domain. The $2F_o\text{-}F_c$ electron density map, contoured to $1\sigma$, is represented by a dark blue mesh. (F) In vitro CDK4/cyclinD1 activity assay in the presence of varying concentrations of ribociclib. The $IC_{50}$ of ribociclib is shown in the inset. Error bars represent the S.E.M. of three replicates. (G, H) Structural comparison between CDK6 and CAMKII bound to inhibitors. Structures of ribociclib and abemaciclib in the active site of CDK6 (G), showing bridging interaction through an ordered water molecule. Ribociclib in CaMKII (H) instead forms a contact with a backbone nitrogen through a water molecule, as seen in Fig. 2. Source data are available online for this figure.

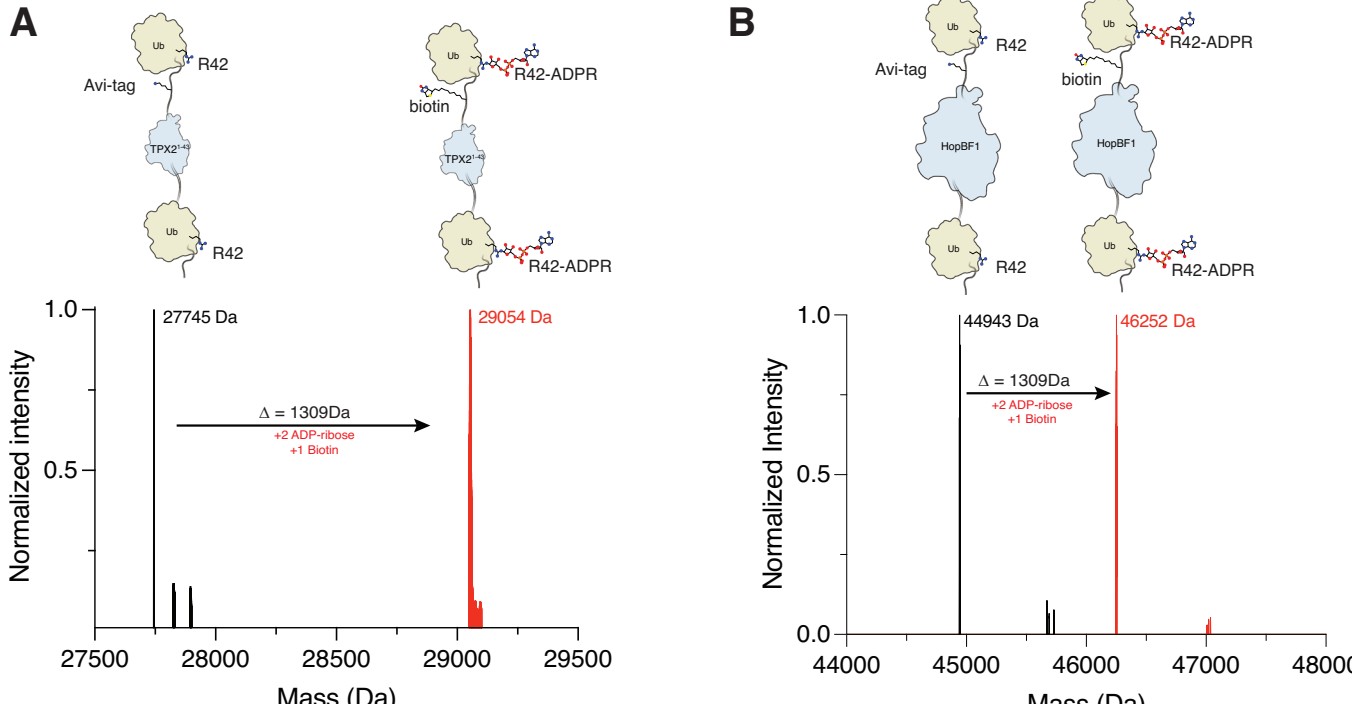

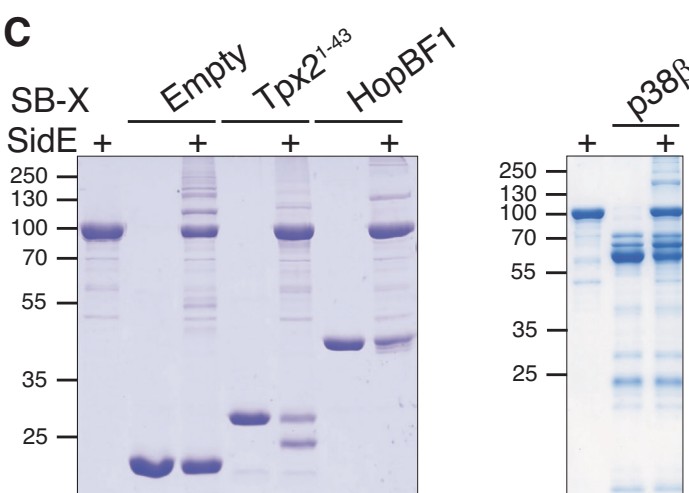

**Figure EV4.   SidBait identifies targets of proteins of interest.**

(A, B) Intact mass spectra of unmodified SidBait-Tpx2$^{1-43}$ (A) and SidBait-HopBF1 (B) (left, black), and the respective SidBait-POI molecules containing a biotin and two ADP-ribose molecules (right, red). (C) NAD$^+$-independent SidE autoubiquitination of the SidBait-POI probes. The probes were incubated with SidE and the reaction products were separated by SDS-PAGE and visualized by Coomassie staining. Source data are available online for this figure.

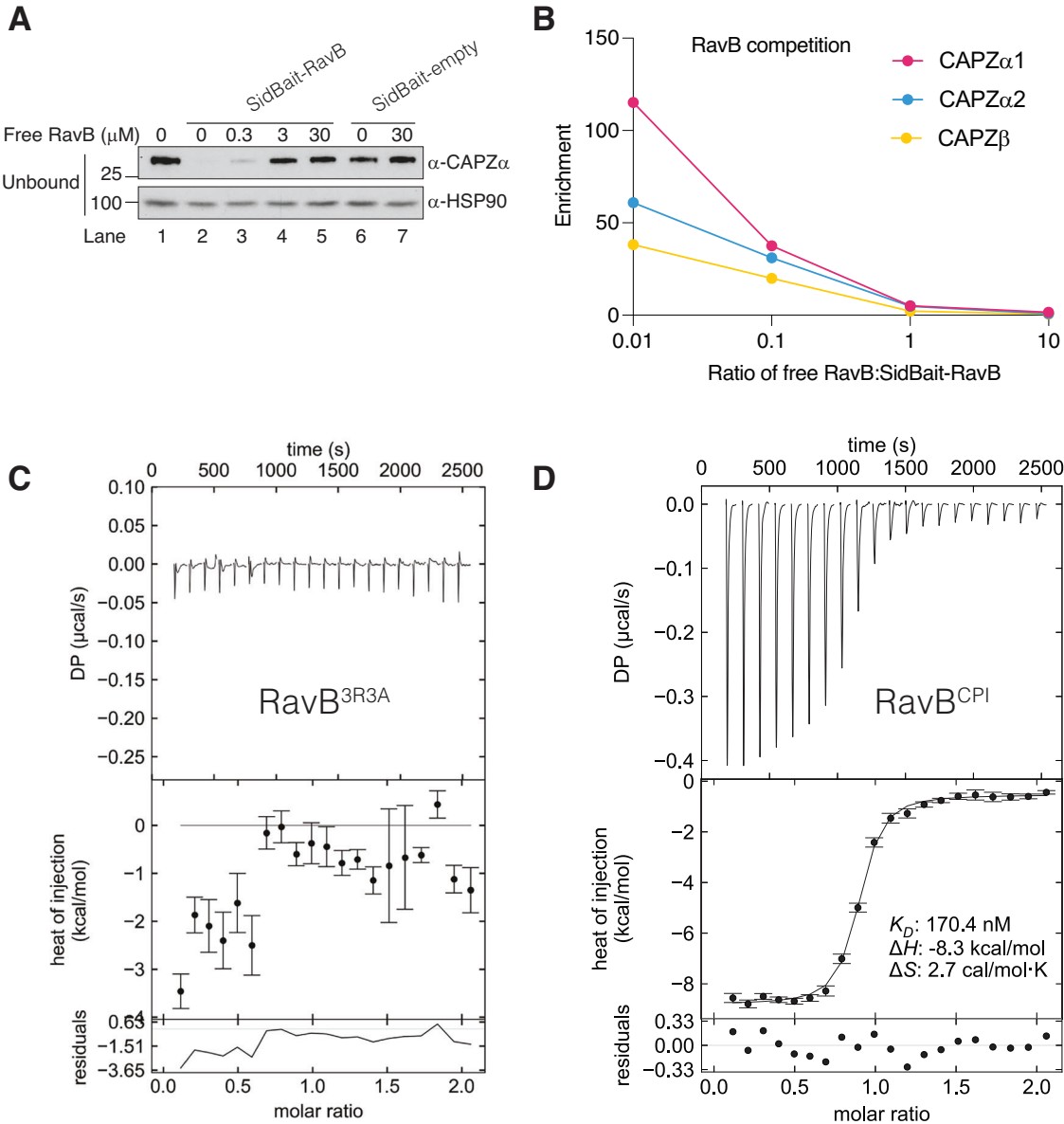

**Figure EV5. SidBait identifies CapZ as a binding partner of the *Legionella* effector RavB.**

(A) Protein immunoblot of the unbound fractions following avidin enrichment of the SidBait-RavB probe from HEK293 cell lysates that have been incubated with and without free RavB. CapZα and HSP90 are shown. CapZ is readily detectable in a cell lysate (lane 1). All cellular CapZ is bound to the SidBait-RavB probe in the absence of free RavB (lane 2). Following the addition of free RavB, the SidBait-RavB probe is competed off RavB (lanes 3–5). (B) Plots of the decreasing fold enrichment of CapZ isoforms from SidBait-RavB probe as quantified by mass spectrometry after addition of increasing competing free RavB. (C, D) Isothermal titration calorimetry (ITC) traces showing the binding of RavB[3R3A] (C) or RavB[108-148] containing the RavB[CPI] (D) to CapZ. In each ITC experiment, the RavB species was injected into the cell containing CapZ. $K_d$, enthalpy and entropy values are shown in the inset. These values are undefined for RavB[3R3A], as no binding was observed. Source data are available online for this figure.

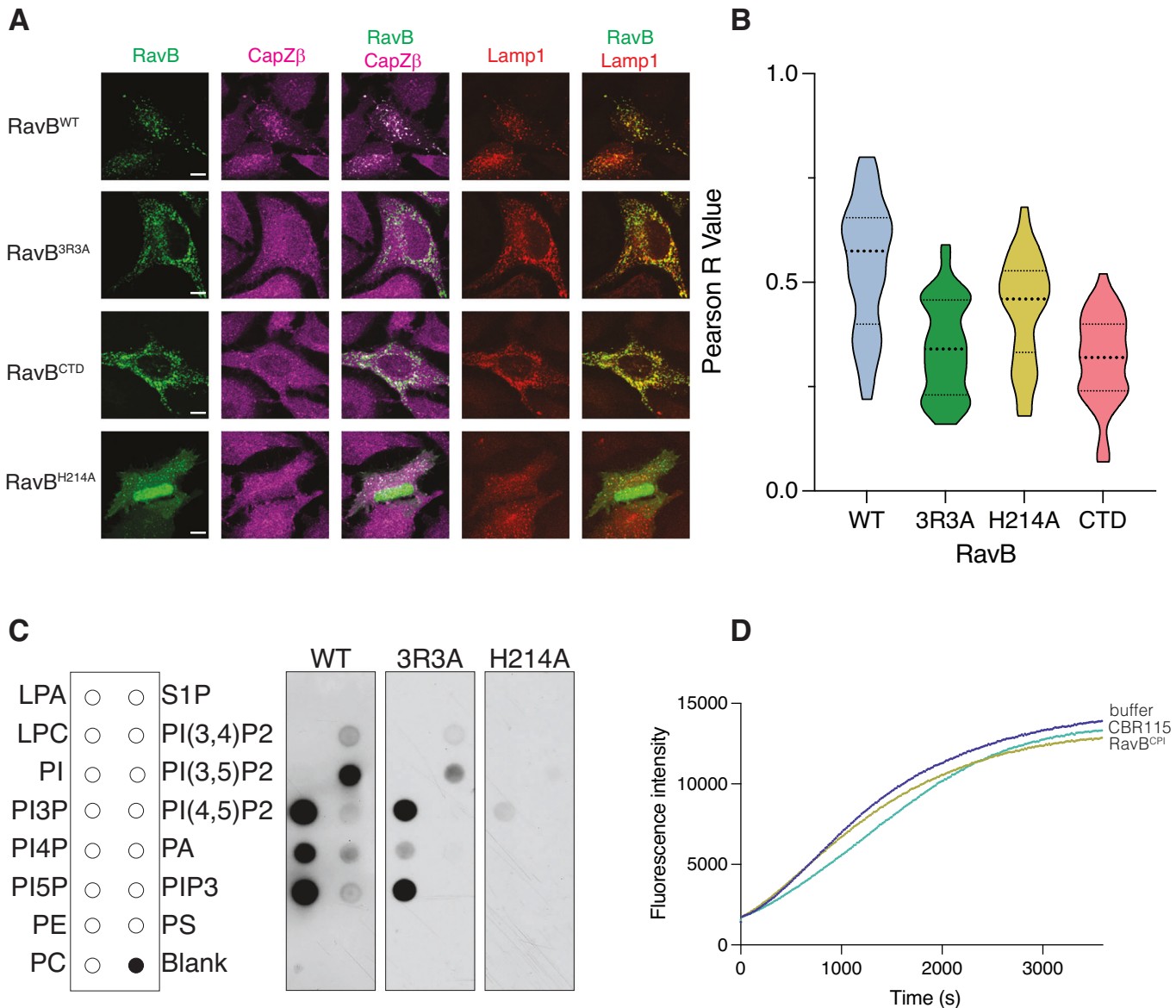

**Figure EV6. RavB is a phosphatidylinositol phosphate binding, actin decapping protein.**

(A) Immunofluorescence microscopy of HeLa cells expressing EGFP-RavB^WT or various mutants, mTagBFP-Lamp1. Endogenous CapZβ is also shown (magenta). The images depict the entire cell from Fig. 5a. Scale bar represents 10 μm. (B) Plot of the Pearson R Value of colocalization between transfected mTagBFP-Lamp1 and endogenous CapZβ in cells expressing WT or various RavB mutants. Each comparison was calculated using 40 cells across 3 independent experiments. (C) Protein immunoblot of RavB or mutants bound to a lipid panel spotted on PIP strips with an anti-RavB antibody. All membranes shown were exposed on the same film. (D) Pyrene-actin polymerization assays demonstrating that the RavB^CPI does not affect actin polymerization. The polymerization of actin was measured in the presence of a buffer control (blue), the known decapping peptide CBR115 (teal) and the RavB^CPI peptide (green). Assays were run with 2 μM G-actin and 1.25 μM decapping peptide. Source data are available online for this figure.

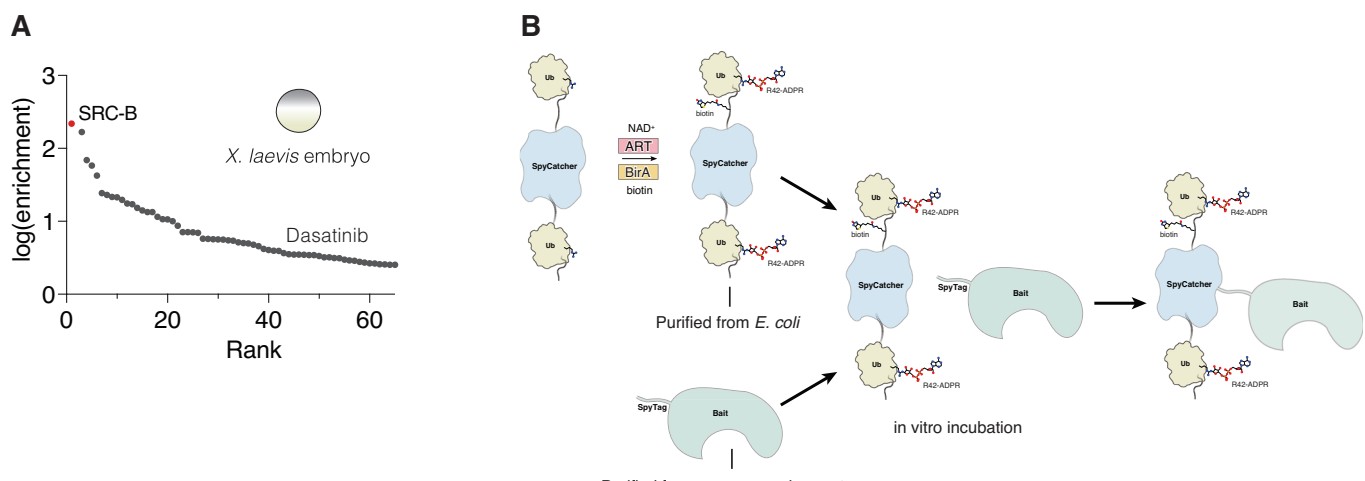

**Figure EV7. Additional applications of SidBait.**

(A) Plot of fold enrichment of proteins from SidBait-dasatinib in live *X. laevis* embryos over pulldowns with the SidBait[C145A] control. (B) Schematic of the SpyTag/SpyCatcher system as applied to SidBait. The Ub-SpyCatcher-Ub fusion protein is co-expressed in *E. coli* with the SidE ART domain and BirA. The ADP-ribosylated and biotinylated protein is incubated with a protein of interest (POI) fused to the SpyTag, which has been purified from an alternative expression system. Upon mixing, the SpyTag-POI spontaneously conjugates to the SpyCatcher protein, forming a stable isopeptide bond. The resulting fusion protein is used in a SidBait experiment.

