## [Peer Review File · The EMBO Journal]

Bacterial ubiquitin ligase engineered for small molecule and protein target identification

James Ye, Abir Majumdar, Brenden Park, Miles Black, Ting-Sung Hsieh, Adam Osinski, Kelly Servage, Kartik Kulkarni, Jacinth Naidoo, Neal Alto, Margaret Stratton, Dominique Alfandari, Joseph Ready, Krzysztof Pawłowski, Diana Tomchick, and Vincent Tagliabracci

Corresponding author(s): Vincent Tagliabracci (Vincent.Tagliabracci@UTSouthwestern.edu)

Review Timeline:

Submission Date:	11th Apr 25
Editorial Decision:	15th May 25
Revision Received:	1st Oct 25
Editorial Decision:	7th Nov 25
Revision Received:	18th Nov 25
Accepted:	21st Nov 25

Editor: William Teale

Transaction Report:

Dear Dr. Tagliabracci,

Thank you again for the submission of your manuscript entitled "Bacterial ubiquitin ligase engineered for small molecule and protein target identification". We have now received the reports from two referees, which I copy below.

As you can see from their comments, both referees were impressed by SidBait, seeing clear and unique potential applications. That said, both of them stress that the manuscript would benefit from a direct comparison with a small selection of currently-popular proximity labelling strategies. As I would envisage this work being published as a 'Resource' article, I agree that such bench-marking is important. There are also a range of more technical points which could mostly be addressed without further experiments.

Based on the overall interest expressed in the reports, I would like to invite you to address the comments of all referees in a revised version of the manuscript. I should add that it is The EMBO Journal policy to allow only a single major round of revision and that it is therefore important to resolve the main concerns at this stage. I believe the concerns of the referees are reasonable and addressable, but please contact me if you have any questions (I am always available for a Zoom call - we can discuss strategies to cover a sensible and informative range of proximity ligation comparisons), need further input on the referee comments or if you anticipate any problems in addressing any of their points. Please, follow the instructions below when preparing your manuscript for resubmission.

I would also like to point out that as a matter of policy, competing manuscripts published during this period will not be taken into consideration in our assessment of the novelty presented by your study ("scooping" protection). We have extended this 'scooping protection policy' beyond the usual 3 month revision timeline to cover the period required for a full revision to address the essential experimental issues. Please contact me if you see a paper with related content published elsewhere to discuss the appropriate course of action.

Again, please contact me at any time during revision if you need any help or have further questions.

Thank you very much again for the opportunity to consider your work for publication. I look forward to your revision.

Best regards,

William

William Teale, Ph.D.
Editor
The EMBO Journal

When submitting your revised manuscript, please carefully review the instructions below and include the following items:

- 1) a .docx formatted version of the manuscript text (including legends for main figures, EV figures and tables). Please make sure that the changes are highlighted to be clearly visible.
- 2) individual production quality figure files as .eps, .tif, .jpg (one file per figure).
- 3) a .docx formatted letter INCLUDING the reviewers' reports and your detailed point-by-point response to their comments. As part of the EMBO Press transparent editorial process, the point-by-point response is part of the Review Process File (RPF), which will be published alongside your paper.
- 4) a complete author checklist, which you can download from our author guidelines ([https://wol-prod-cdn.literatumonline.com/pb-assets/embo-site/Author Checklist%20-%20EMBO%20J-1561436015657.xlsx](https://wol-prod-cdn.literatumonline.com/pb-assets/embo-site/Author%20Checklist%20-%20EMBO%20J-1561436015657.xlsx)). Please insert information in the checklist that is also reflected in the manuscript. The completed author checklist will also be part of the RPF.
- 5) Please note that all corresponding authors are required to supply an ORCID ID for their name upon submission of a revised manuscript.
- 6) We require a 'Data Availability' section after the Materials and Methods. Before submitting your revision, primary datasets

produced in this study need to be deposited in an appropriate public database, and the accession numbers and database listed under 'Data Availability'. Please remember to provide a reviewer password if the datasets are not yet public (see <https://www.embopress.org/page/journal/14602075/authorguide#datadeposition>). If no data deposition in external databases is needed for this paper, please then state in this section: This study includes no data deposited in external repositories. Note that the Data Availability Section is restricted to new primary data that are part of this study.

Note - All links should resolve to a page where the data can be accessed.

8) For data quantification: please specify the name of the statistical test used to generate error bars and P values, the number (n) of independent experiments (specify technical or biological replicates) underlying each data point and the test used to calculate p-values in each figure legend. The figure legends should contain a basic description of n, P and the test applied. Graphs must include a description of the bars and the error bars (s.d., s.e.m.).

9) We would also encourage you to include the source data for figure panels that show essential data. Numerical data can be provided as individual .xls or .csv files (including a tab describing the data). For 'blots' or microscopy, uncropped images should be submitted (using a zip archive or a single pdf per main figure if multiple images need to be supplied for one panel). Additional information on source data and instruction on how to label the files are available at .

10) We replaced Supplementary Information with Expanded View (EV) Figures and Tables that are collapsible/expandable online (see examples in <https://www.embopress.org/doi/10.15252/embj.201695874>). A maximum of 5 EV Figures can be typeset. EV Figures should be cited as 'Figure EV1, Figure EV2" etc. in the text and their respective legends should be included in the main text after the legends of regular figures.

12) Our journal encourages inclusion of *data citations in the reference list* to directly cite datasets that were re-used and obtained from public databases. Data citations in the article text are distinct from normal bibliographical citations and should directly link to the database records from which the data can be accessed. In the main text, data citations are formatted as follows: "Data ref: Smith et al, 2001" or "Data ref: NCBI Sequence Read Archive PRJNA342805, 2017". In the Reference list, data citations must be labeled with "[DATASET]". A data reference must provide the database name, accession number/identifiers and a resolvable link to the landing page from which the data can be accessed at the end of the reference. Further instructions are available at .

13) In order to increase the reproducibility and reach of your work, The EMBO Journal includes a table of reagents that were used in the study. Please provide this along with your revisions.

Futher instructions for preparing your revised manuscript:

When assembling figures, please refer to our figure preparation guideline in order to ensure proper formatting and readability in

print as well as on screen:

We realize that it is difficult to revise to a specific deadline. In the interest of protecting the conceptual advance provided by the work, we recommend a revision within 3 months (13th Aug 2025). Please discuss the revision progress ahead of this time with the editor if you require more time to complete the revisions. Use the link below to submit your revision:

Referee #1:

Ye et al., report a proximity labelling method called SidBait that is based on the novel ubiquitin ligase from *Legionella pneumophila* (LP). SidBait exploits SidE family enzymes from LP which catalyze ADP-ribosylation and subsequent phosphoribosyl ubiquitination of target proteins. They use a SNAP tag fused on both sides to a copy of ubiquitin molecule and express this moiety in *E. coli* in the presence of BirA and mono-ADP ribosyl transfer domain from SidEs. The fully modified (ADP-ribosylated and biotinylated) form of this protein is used as a bait to incubate with cell lysates to trap the target through Phosphodiesterase domain mediated ubiquitination. For small molecule target identification, SidBait requires the fusion of the molecule of interest to chloropyrimidine (CLP) which covalently attaches to SNAP and allows covalent entrapment of the target directly in the cell lysate. For mapping protein-protein interactions through SidBait, authors replace the SNAP tag with the protein of interest. Overall, the method seems to work at least in the examples shown, however without comparison to one or more widely used methods already existing, it is hard to understand how this method will be positioned among the myriad of other methods out there. Other major comments are listed below.

Major comments:

- 1) The target identification of small molecules is a really interesting application of SidBait in principle. However, what I see as the most important aspect for this to work is covalent attachment of the small molecule of interest to the chloropyrimidine (CLP) moiety. I am assuming that the authors are aware of the binding mode of each of the small molecule (investigated here) to the target and therefore carefully chose the precise way to fuse the small molecule with CLP so that this fusion does not interfere with the expected function of the small molecule. What if this fusion of small molecule and CLP prevents the small molecule from capturing some atypical/uncharacterized interactions through a novel binding mode? Can the others explain how to go about this for a small molecule whose mode of action is unknown? Or state that this method would be most useful for already well characterized ligands whose secondary targets may be trapped here. In any case, this aspect should be discussed and limitations should be underlined.
- 2) Global thermal proteome profiling and other methods that authors cite are available to map the targets of small molecules, it would be important to compare this method to at least one of the leading methods in the field and provide comparison.
- 3) Can the authors explain why they include two ubiquitin molecules in SidBait, because in principle one seems to be enough for

the method to function?

4) It would be a stretch to say that SidE PDE domains do not have sequence specificity. In fact, Kalayil et al., 2018 that the authors cite maps the consensus sequence of PDE substrates. In the limited tests, it was revealed that some sequences around the target serine are incompatible with ubiquitination. Therefore, in the context of SidBait, targets which have these consensus motifs readily available to the PDE might get modified to a greater extent compared to the tighter binders which have no consensus sequence for phosphoribosyl ubiquitination. This might also explain the ribociclib experiment when the authors found CamKII first but upon longer incubation detected CDK4. Although authors try to explain the preferred selectivity of ribociclib towards CaMKII over CDK4 using the active site residue mutations, this might just be one of the factors. It would be important to have a comparison of the binding affinities of this molecule to both these kinases.

5) For the application of SidBait to proteins, it is important to compare this method to other proximity labelling techniques. There are a number of them but comparison to TurboID would be a good start since this is the most widely used now a days.

6) It is a little surprising that the bona fide substrates of SidEs in cells are not picked up here. Can the authors comment on whether Rtn4 or Rab33 are found commonly in the list of targets here?

7) From the small molecule interactions and protein-protein interactions mapped here only p38 interactions seems to be really transient. The rest are very high to high affinity binders. For eg. can the interaction between RavB and CapZ be mapped simply by IP-MS method? The method needs to be tested on more transient interactions and compared to other methods such as TurboID to convince that it maps transient interactions.

Referee #2:

Here the authors have exploited their deep knowledge of the effector systems of the *Legionella pneumophila* intracellular bacterial pathogen to develop a novel modular proximity ligation detector system, called SidBait, which is based on the *Legionella* SidE effector system that ubiquitylates host proteins, independently of the host E1-E2-E3 Ub cascade, attaching Ub to Ser or Tyr residues on host proteins through a phosphoribose linkage. In Step 1 of SidBait, the two Ub moieties of a 6xHis-Ub-Avi-SNAP-Ub fusion protein are ADP-ribosylated in *E. coli* by co-expression with the SidE ART domain and the Avi target motif is biotinylated by co-expressed BirA. The modified SidBait proteins are purified using Ni-NTA affinity chromatography. In Step 2, the SNAP domain in purified SidBait is modified with a chloropyrimidine (CLP) derivative of a small molecule of interest and then the modified 6xHis-Ub-Avi-SNAP-Ub fusion protein is added to a cell/tissue lysate together with the purified SidE PDE domain, whereupon proteins that bind to the small molecule are biotin-ubiquitylated by the SidE PDE, allowing them to be affinity purified with avidin and then identified by MS analysis. In a variant strategy, this schema is used to identify binding partners of a protein of interest (POI), by replacing the SNAP moiety with a POI, this time allowing proteins that interact tightly with the POI in a lysate to be identified. Using Scheme 1, the authors analyzed the interactomes of alisertib, an AurA kinase inhibitor, BI2536, a PLK inhibitor, and dasatinib, a multi-tyrosine kinase inhibitor, in each case identifying the target kinase among the most frequent interactors, as known off target hits, such as BRD2-4 for BI2536, and ACD19 for alisertib. A similar analysis for the CDK4/6 inhibitor ribociclib, a drug approved for breast cancer therapy, surprisingly led to the identification of CaMKII β and CaMKII δ as the most prominent hits, which were shown to be potentially inhibited by ribociclib in an in vitro kinase assay (IC₅₀=150 nM; this compares to ribociclib's Cdc2/CDK4 IC₅₀ = 10 nM). They also solved a crystal structure of the kinase domain of CaMKII δ bound to ribociclib demonstrating its specific binding mode in the ATP-binding pocket. Consistent with its ability to inhibit CaMKII, ribociclib inhibited beating of cultured cardiomyocytes, a CaMKII dependent process. Using Scheme 2, they assessed the interactomes of SidBaits containing the Tpx2 (Tpx21-43) Aurora A-activating segment, HopBF1, a bacterial effector kinase that phosphorylates eukaryotic the HSP90 chaperone, and the p38 β MAPK (MAPK11), finding AurA, Hsp90 α/β , and NCK1/MAP2K4/CSK/S6K as the top hits respectively. Finally, they used SidBait to identify interactors for the uncharacterized *Legionella* RavB effector protein, which is predicted to have a helical N-terminal segment and a PIPx-binding C-terminal domain connected by a long unstructured linker. SidBait-RavB identified the CapZ α and CapZ β subunits of the CapZ F-actin capping complex. A 2 Å crystal structure of the RavB:CapZ complex revealed how RavB binds with high affinity to CapZ via a 17 aa conserved CPI motif to form a ternary complex. Mutational analysis of the PIPx-binding C-terminal domain showed that PIPx binding was required for localization of RavB to the lysosomal membrane. A complex of CapZ with the RavB CPI peptide exhibited higher F-actin decapping activity than CapZ suggesting that RavB acts as an allosteric activator of CapZ decapping activity, at the lysosomal membrane. Finally, they demonstrated the versatility of the SidBait approach, showing that both small molecule and protein baits could be used to identify known target proteins in mouse heart/kidney tissue extracts, yeast extracts, and also when microinjected into fertilized *Xenopus* eggs, which were allowed to develop into embryos.

The SidBait modular proximity ligation system the authors have devised is a powerful new molecular interaction detection method, but it is a little complicated to employ, and has the limitation that it can only be used in cell/tissue lysates or cells that are large enough to be microinjected. The authors argue that this feature is a strength of the SidBait system because it allows one to explore compartments not accessible in the cell, but it is something of a limitation compared to proximity systems that can be expressed genetically and localized in the proper subcellular compartment. Nevertheless, the authors have demonstrated convincingly that SidBait works well to identify binding partners for small molecules and for proteins of interest, validating with known controls, and the three examples they studied have revealed some interesting functional insights as well small molecule kinase inhibitor off-target interactions.

Overall, the paper seems a little disjointed, because it includes three different examples of SidBait use in three different contexts

that were explored to different extents. In some ways, SidBait's ability to uncover potential off-target binders of small molecules, such as the clinically approved ribociclib CDK4 inhibitor, is potentially the most useful. Indeed, the strong binding of the clinically-approved CDK4 inhibitor ribociclib to CaMKII and the evidence that it can inhibit CaMKII in vitro and in cells uncovers an unexpected off-target effect of ribociclib, which, as they indicate, could be relevant to the rare cardiac side effects observed with ribociclib. In fact, given its 150 nM potency towards CaMKII δ , it is surprising that this off-target effect of ribociclib has not been reported previously (see point 5). The discovery that the previously uncharacterized Legionella RavB protein acts as a phospholipid-binding F-actin decapping protein activator is a nice story in and of itself, and includes detailed mechanistic insights with co-crystal structures as an added bonus.

In summary, in one sense this paper belongs in a methods journal, but it describes an ingenious and novel proximity ligation method with enough interesting functional insights that extends it beyond a simple method description.

Points: 1. Figure 1: BI2536 is a dual PLK1/bromodomain inhibitor, but while its binding mode to PLK1 is known, how does it bind to BRD4, and is it clear that the linker-CLIP moiety would not interfere with binding?

2. Figure 2: Surprisingly, CDK4/6 (and cyclin D), ribociclib's intended target were not identified as frequent binders. The authors suggest that this may be due to the inability of CDK4 to bind ATP-competitive inhibitors, such as ribociclib, when it is in the heterotrimeric CDK4:CyclinD:p27 complex state, and that a longer incubation with the SidBait-ribociclib bait might be needed to capture CDK4 when it dissociates. In this connection, they showed that incubation of SidBait-ribociclib in a cell lysate for 72 hours prior to the addition of the SidE PDE domain did result in the enrichment of CDK4 (and cyclin D?), but a SidBait-ribociclib/CDK4 complex seem unlikely to be stable enough to accumulate (K_d for CDK4 = 380 nM and for CycD1/CDK4 = 1.5 nM), and one would need SidE-PDE to be present continuously rather than being added for the last hour. Ribociclib binds CycD/CDK4 complexes more avidly than free CDK4, but the linker-CLP moiety might interfere with efficient binding of Ribociclib-CLP to CycD1/CDK4 (does ribociclib-CLP inhibit CycD1/CDK4 activity with the same potency as ribociclib?), which may be the major species of CDK4 in the lysate, and the lag in binding might be dependent on CycD degradation over 72 hours. Ideally, the authors need to determine the status of CDK4 complexes in the lysates they used, i.e. is CDK4/CycD/p27 the major species in confluent HEK293A cells?. Biochemical experiments testing the efficiencies of SidBait-ribociclib labeling of purified CycD1/CDK4 and CycD1/CDK4/p27 complexes versus free CDK4 could also be informative. If CDK4 in CycD/CDK4 complexes is Ub-biotinylated, is CycD also labeled?

3. Figure 2C: The legend should indicate that the CaMKII δ isoform was used in this assay. Thr286 is a CaMKII autophosphorylation site, whose phosphorylation occurs upon Ca²⁺/Ca²⁺ stimulation of one subunit in the holoenzyme leading to phosphorylation of the Thr286 residue on a neighboring subunit. Thus, while T286 phosphorylation is a measure of CaMKII kinase activity, it is a special case because the substrate is captive and has unusual phosphorylation kinetics. Presumably, the purified CaMKII δ fragment used for these experiments was a dimer, but the same "intramolecular" phosphorylation scenario applies, and, for this reason, it might be better to measure phosphorylation of an added CaMKII substrate to determine ribociclib's IC₅₀. as was done for measurement of the potency CycD1/CDK4 inhibition (Figure S2e). The IC₅₀ for CycD1/CDK4 was 2 nM. i.e., 75 times more potent than towards CaMKII.

4. Figure 2C: In the cardiomyocyte experiments, did the authors show that ribociclib reduced phosphorylation of known CaMKII sites in its target proteins, as well that of pT268?

5. With regard to CaMKII as a target for ribociclib, the Novartis group's in depth comparison of the specificities of ribociclib, palbociclib, and abemaciclib (Kim et al. Oncotarget 9:35226-35240, 2018) showed that of the ~400 kinases tested in a specificity screen CaMKII family members were the second most susceptible kinases to ribociclib inhibition after CDK4/6 - but at 100 nM ribociclib only 30% CaMKII inhibition was observed.

6. Figure 3: The authors could expand their comments on some of the observed interactors in panels B-D. For instance, the fact that the HopBF1 bait interacted most efficiently with the LTN1/listerin E3 Ub ligase, but it is not clear if LTN1 is this a known interactor. The p38 β MAP kinase interacted most strongly with the NCK1 adaptor protein, which they indicate is a known p38 β interactor. However, with regard to NCK1 being a p38 β substrate, Kinase Library predictions rank p38 β as the 29th most likely kinase to phosphorylate S91, the 47th most likely kinase to phosphorylate S166 and the 48th most likely to phosphorylate S262, the three reported Ser/Thr.Pro sites in NCK1, and p38 β is not listed among the 216 NCK1 interactors in BioGRID. Also, CSK, another known p38 β interactor does not seem to have any known Ser/Thr.Pro phosphosites, and as a dedicated SFK kinase it seems unlikely that CSK would phosphorylate p38 β . In addition, p38 β is not listed among the 871 CSK interactors in BioGRID.

7. Figure 6: With regard to additional applications of SidBait, have the authors tested whether, with modifications to allow RNA tagging, SidBait could also be used to identify sequence-specific RNA-binding proteins?

8. Page 8/9: Although the authors argue that SidBait's use in cell/tissue lysates is a virtue, this could result in detection of spurious interactions due to loss of compartment boundaries, for instance upon nuclear and organelle lysis.

9 From a SidBait design perspective, would the inclusion and use of (Tev) protease cleavage sites in the Ub-bait linker beyond the Avi-tag biotinylation site and close to the bait protein give cleaner interactome results?

We thank the referees for their comments and suggestions. We have addressed them below:

Referee #1:

Ye et al., report a proximity labelling method called SidBait that is based on the novel ubiquitin ligase from Legionella pneumophila (LP). SidBait exploits SidE family enzymes from LP which catalyze ADP-ribosylation and subsequent phosphoribosyl ubiquitination of target proteins. They use a SNAP tag fused on both sides to a copy of ubiquitin molecule and express this moiety in E.coli in the presence of BirA and mono-ADP-ribosyl transfer domain from SidEs. The fully modified (ADP-ribosylated and biotinylated) form of this protein is used as a bait to incubate with cell lysates to trap the target through Phosphodiesterase domain mediated ubiquitination. For small molecule target identification, SidBait requires the fusion of the molecule of interest to chloropyrimidine (CLP) which covalently attaches to SNAP and allows covalent entrapment of the target directly in the cell lysate. For mapping protein-protein interactions through SidBait, authors replace the SNAP tag with the protein of interest. Overall, the method seems to work at least in the examples shown, however without comparison to one or more widely used methods already existing, it is hard to understand how this method will be positioned among the myriad of other methods out there. Other major comments are listed below.

Major comments:

1) The target identification of small molecules is a really interesting application of SidBait in principle. However, what I see as the most important aspect for this to work is covalent attachment of the small molecule of interest to the chloropyrimidine (CLP) moiety. I am assuming that the authors are aware of the binding mode of each of the small molecule (investigated here) to the target and therefore carefully chose the precise way to fuse the small molecule with CLP so that this fusion does not interfere with the expected function of the small molecule. What if this fusion of small molecule and CLP prevents the small molecule from capturing some atypical/uncharacterized interactions through a novel binding mode? Can the others explain how to go about this for a small molecule whose mode of action is unknown? Or state that this method would be most useful for already well characterized ligands whose secondary targets may be

trapped here. In any case, this aspect should be discussed and limitations should be underlined.

The positioning of the CLP moiety is an important consideration in small molecule SidBait experiments, and our choice of CLP-placement was indeed informed by the known binding mode of each inhibitor. In the case of a small molecule with no known binding mode, we envision using structure-activity relationship (SAR) data to best position the CLP, or, as a last resort, synthesizing multiple probes with the CLP moiety protruding from different points of the molecule. However, we recognize that this would be substantially more challenging than our proof-of-concept experiments and, furthermore, that there may be small molecules for which it is simply not possible to append a CLP handle without interfering with binding. We have amended the discussion to include these points.

2) Global thermal proteome profiling and other methods that authors cite are available to map the targets of small molecules, it would be important to compare this method to at least one of the leading methods in the field and provide comparison.

We provide comparisons of SidBait experiments conducted using the four inhibitors we had originally presented in the manuscript, this time in K562 cell lysates against the results reported in van Vranken et al. 2024 (PMID: 39526730). Importantly, using SidBait, we pick up targets not detected using the PISA method (CaMKII for ribociclib, ACAD10 for alisertib, several tyrosine kinases and SIK2 for dasatinib). See **Figure R1** (Supplementary Fig. 2 in the manuscript) below:

Figure R1: Comparison of small molecule target identification between thermal shift assay (PISA; van Vranken *et al.* 2024; PMID: 39526730) and SidBait in K562 cells. (a, b) Fold change plots of the enrichment of targets for alisertib using the PISA method (a), and the SidBait method (b). (c, d) Fold change plots of the enrichment of targets for BI2536 using the PISA method (c), and the SidBait method (d). (e, f) Fold change plots of the enrichment of targets for dasatinib using the PISA method (e), and the SidBait method (f). (g, h) Fold change plots of the enrichment of targets for ribociclib using the PISA method (g), and the SidBait method (h). Red dots represent known targets.

3) *Can the authors explain why they include two ubiquitin molecules in SidBait, because in principle one seems to be enough for the method to function?*

The reviewer is correct in that one ubiquitin is sufficient for SidBait. The second ubiquitin improves the probability of target capture by 1) increasing the effective radius of the bait and 2) providing a second opportunity for the covalent linkage to the target to form, since conversion of ADPR to pR by the PDE domain in the absence of a substrate is irreversible.

4) *It would be a stretch to say that SidE PDE domains do not have sequence specificity. In fact, Kalayil et al., 2018 that the authors cite maps the consensus sequence of PDE substrates. In the limited tests, it was revealed that some sequences around the target serine are incompatible with ubiquitination. Therefore, in the context of SidBait, targets which have these consensus motifs readily available to the PDE might get modified to a greater extent compared to the tighter binders which have no consensus sequence for phosphoribosyl ubiquitination.*

Although there seems to be some sequence specificity for the SidE PDE domain in a *cellular context*, the sequence logo in Figure 3d from Kalayil et al. 2018 (PMID: 29795347) (**Figure R2**) demonstrates that the most important substrate determinant for pR-ubiquitination is the presence of a serine. We have made a minor update to the language in the main text to reflect this (“The SidE ligases do not require a **strong** consensus sequence...”). We note that another study suggests that SidE PDE domain

Figure R2. The SidE Ligase have limited sequence specificity. Sequence logo for the substrate specificity for the SidE ligases. From Kalayil *et al.* (2018); PMID: 29795347.

can also target tyrosines. (Zhang et al. 2021; PMID 34704056).

This might also explain the ribociclib experiment when the authors found CamKII first but upon longer incubation detected CDK4. Although authors try to explain the preferred selectivity of ribociclib towards CaMKII over CDK4 using the active site residue mutations, this might just be one of the factors. It would be important to have a comparison of the binding affinities of this molecule to both these kinases.

We did not intend to give the impression that ribociclib is more selective for CaMKII than CDK4, and in fact our explanation invoking the active site residues argues for the opposite—that ribociclib has far greater potency towards CDK4/6 than CaMKII. It is also possible that the relative abundance of the target proteins is an important determinant of targets we detect in our experiments. For example, in K562 cells, the SidBait-ribociclib experiments pick up CDK6 instead of CDK4. We have included a note about this in the discussion.

5) For the application of SidBait to proteins, it is important to compare this method to other proximity labelling techniques. There are a number of them but comparison to TurboID would be a good start since this is the most widely used now a days.

Although we appreciate that some form of benchmarking may appear useful to users considering their options, we believe that the protein space is too varied to perform a meaningful comparison across techniques. For example, we were unable to enrich the CapZ complex using TurboID-RavB, although we are not sure why, as we expected this experiment to produce similar results to our SidBait experiments. The intent of this work is to add another tool to the molecular biology toolbox, showcase its capabilities and demonstrate how to follow up on the hypotheses generated through its use, as we have done in the case of a previously uncharacterized bacterial effector. We expect that SidBait will perform better for some applications while published proximity ligation techniques will be better for others, but we believe that conducting a series of experiments with a wide enough range of baits to satisfy this question seems outside of the scope of this work.

6) It is a little surprising that the bona fide substrates of SidEs in cells are not picked up

here. Can the authors comment on whether Rtn4 or Rab33 are found commonly in the list of targets here?

We often see Rtn4 in our SidBait experiments (see source data tables); however, since the bait and control are closely matched, Rtn4 appears in *both* samples and therefore washes out in the enrichment calculation.

7) From the small molecule interactions and protein-protein interactions mapped here only p38 interactions seems to be really transient. The rest are very high to high affinity binders. For eg. can the interaction between RavB and CapZ be mapped simply by IP-MS method? The method needs to be tested on more transient interactions and compared to other methods such as TurboID to convince that it maps transient interactions.

While we don't expect SidBait to capture every transient interaction, we do have several pieces of evidence to suggest that SidBait can capture transient interactions. First, our data with p38 β demonstrates that we can enrich the upstream kinase, MAP2K4, which interacts with p38 β in the high micromolar range. Second, we can enrich p38 β substrates, which—to the best of our knowledge—do not form stable complexes with p38 β . Third, the HSP90 kinase HopBF1 does not stably interact with HSP90 in vitro and has a K_M of $\sim 3 \mu\text{M}$ for HSP90. Using SidBait-HopBF1, we successfully enriched HSP90 from HEK293 cells. While it is likely that the interaction between RavB and CapZ could be detected by IP-MS, the covalent linkage introduced by SidBait allows for stringent washing of the beads, yielding a much cleaner result.

Referee #2:

Here the authors have exploited their deep knowledge of the effector systems of the Legionella pneumophila intracellular bacterial pathogen to develop a novel modular proximity ligation detector system, called SidBait, which is based on the Legionella SidE effector system that ubiquitylates host proteins, independently of the host E1-E2-E3 Ub cascade, attaching Ub to Ser or Tyr residues on host proteins through a phosphoribose

linkage. In Step 1 of SidBait, the two Ub moieties of a 6xHis-Ub-Avi-SNAP-Ub fusion protein are ADP-ribosylated in *E. coli* by co-expression with the SidE ART domain and the Avi target motif is biotinylated by co-expressed BirA. The modified SidBait proteins are purified using Ni-NTA affinity chromatography. In Step 2, the SNAP domain in purified SidBait is modified with a chloropyrimidine (CLP) derivative of a small molecule of interest and then the modified 6xHis-Ub-Avi-SNAP-Ub fusion protein is added to a cell/tissue lysate together with the purified SidE PDE domain, whereupon proteins that bind to the small molecule are biotin-ubiquitylated by the SidE PDE, allowing them to be affinity purified with avidin and then identified by MS analysis. In a variant strategy, this schema is used to identify binding partners of a protein of interest (POI), by replacing the SNAP moiety with a POI, this time allowing proteins that interact tightly with the POI in a lysate to be identified. Using Scheme 1, the authors analyzed the interactomes of alisertib, an AurA kinase inhibitor, BI2536, a PLK inhibitor, and dasatinib, a multi-tyrosine kinase inhibitor, in each case identifying the target kinase among the most frequent interactors, as known off target hits, such as BRD2-4 for BI2536, and ACD19 for alisertib. A similar analysis for the CDK4/6 inhibitor ribociclib, a drug approved for breast cancer therapy, surprisingly led to the identification of CaMKII β and CaMKII δ as the most prominent hits, which were shown to be potently inhibited by ribociclib in an in vitro kinase assay (IC₅₀=150 nM; this compares to ribociclib's CycD/CDK4 IC₅₀ = 10 nM). They also solved a crystal structure of the kinase domain of CaMKII δ bound to ribociclib demonstrating its specific binding mode in the ATP-binding pocket. Consistent with its ability to inhibit CaMKII, ribociclib inhibited beating of cultured cardiomyocytes, a CaMKII dependent process. Using Scheme 2, they assessed the interactomes of SidBaits containing the Tpx2 (Tpx2 1-43) Aurora A-activating segment, HopBF1, a bacterial effector kinase that phosphorylates eukaryotic the HSP90 chaperone, and the p38 β MAPK (MAPK11), finding AurA, Hsp90 α/β , and NCK1/MAP2K4/CSK/S6K as the top hits respectively. Finally, they used SidBait to identify interactors for the uncharacterized Legionella RavB effector protein, which is predicted to have a helical N-terminal segment and a PIPx-binding C-terminal domain connected by a long unstructured linker. SidBait-RavB identified the CapZ α and CapZ β subunits of the CapZ F-actin capping complex. A 2 Å crystal structure of the RavB:CapZ complex revealed

how RavB binds with high affinity to CapZ via a 17 aa conserved CPI motif to form a ternary complex. Mutational analysis of the PIPx-binding C-terminal domain showed that PIPx binding was required for localization of RavB to the lysosomal membrane. A complex of CapZ with the RavB CPI peptide exhibited higher F-actin decapping activity than CapZ suggesting that RavB acts as an allosteric activator of CapZ decapping activity, at the lysosomal membrane. Finally, they demonstrated the versatility of the SidBait approach, showing that both small molecule and protein baits could be used to identify known target proteins in mouse heart/kidney tissue extracts, yeast extracts, and also when microinjected into fertilized *Xenopus* eggs, which were allowed to develop into embryos.

The SidBait modular proximity ligation system the authors have devised is a powerful new molecular interaction detection method, but it is a little complicated to employ, and has the limitation that it can only be used in cell/tissue lysates or cells that are large enough to be microinjected. The authors argue that this feature is a strength of the SidBait system because it allows one to explore compartments not accessible in the cell, but it is something of a limitation compared to proximity systems that can be expressed genetically and localized in the proper subcellular compartment. Nevertheless, the authors have demonstrated convincingly that SidBait works well to identify binding partners for small molecules and for proteins of interest, validating with known controls, and the three examples they studied have revealed some interesting functional insights as well small molecule kinase inhibitor off-target interactions.

Overall, the paper seems a little disjointed, because it includes three different examples of SidBait use in three different contexts that were explored to different extents. In some ways, SidBait's ability to uncover potential off-target binders of small molecules, such as the clinically approved ribociclib CDK4 inhibitor, is potentially the most useful. Indeed, the strong binding of the clinically-approved CDK4 inhibitor ribociclib to CaMKII and the evidence that it can inhibit CaMKII *in vitro* and in cells uncovers an unexpected off-target effect of ribociclib, which, as they indicate, could be relevant to the rare cardiac side effects observed with ribociclib. In fact, given its 150 nM potency towards CaMKII δ ,

it is surprising that this off-target effect of ribociclib has not been reported previously (see point 5). The discovery that the previously uncharacterized Legionella RavB protein acts as a phospholipid-binding F-actin decapping protein activator is a nice story in and of itself, and includes detailed mechanistic insights with co-crystal structures as an added bonus.

In summary, in one sense this paper belongs in a methods journal, but it describes an ingenious and novel proximity ligation method with enough interesting functional insights that extends it beyond a simple method description.

Points:

1. Figure 1: BI2536 is a dual PLK1/bromodomain inhibitor, but while its binding mode to PLK1 is known, how does it bind to BRD4, and is it clear that the linker-CLIP moiety would not interfere with binding?

BI2536 has been shown to bind the acetyllysine binding site of BRD4 by the Schonbrunn and Knapp labs (PMID: 24568369, 24584101). Given that BRD4 is strongly enriched in our results, we believe that the CLP handle does not interfere with binding. We were lucky in this regard. We have added a note to the discussion about the positioning of the CLP handle.

2. Figure 2: Surprisingly, CDK4/6 (and cyclin D), ribociclib's intended target were not identified as frequent binders. The authors suggest that this may be due to the inability of CDK4 to bind ATP-competitive inhibitors, such as ribociclib, when it is in the heterotrimeric CDK4:CyclinD:p27 complex state, and that a longer incubation with the SidBait-ribociclib bait might be needed to capture CDK4 when it dissociates. In this connection, they showed that incubation of SidBait-ribociclib in a cell lysate for 72 hours prior to the addition of the SidE PDE domain did result in the enrichment of CDK4 (and cyclin D?), but a SidBait-ribociclib/CDK4 complex seem unlikely to be stable enough to accumulate (K_d for CDK4 = 380 nM and for CycD1/CDK4 = 1.5 nM), and one would need SidE-PDE to be present continuously rather than being added for the last hour. Ribociclib binds CycD/CDK4 complexes more avidly than free CDK4, but the linker-CLP moiety might interfere with efficient binding of Ribociclib-CLP to CycD1/CDK4 (does

ribociclib-CLP inhibit CycD1/CDK4 activity with the same potency as ribociclib?), which may be the major species of CDK4 in the lysate, and the lag in binding might be dependent on CycD degradation over 72 hours. Ideally, the authors need to determine the status of CDK4 complexes in the lysates they used, i.e. is CDK4/CycD/p27 the major species in confluent HEK293A cells? Biochemical experiments testing the efficiencies of SidBait-ribociclib labeling of purified CycD1/CDK4 and CycD1/CDK4/p27 complexes versus free CDK4 could also be informative. If CDK4 in CycD/CDK4 complexes is Ub-biotinylated, is CycD also labeled?

We appreciate that inhibitor binding to the CDK4/6:cyclinD complexes is more nuanced than a simple drug-protein interaction. Although the proposed experiments would be informative, we believe that a detailed study is outside the scope of this paper and would distract from the key takeaways, i.e. SidBait can be used to uncover unexpected targets of small molecules. We note that in our subsequent experiments using K562 cell lysates, we were able to pick up CDK6 as a target for ribociclib. This is now shown in Supplementary Fig. 2h.

3. Figure 2C: The legend should indicate that the CaMKII δ isoform was used in this assay. Thr286 is a CaMKII autophosphorylation site, whose phosphorylation occurs upon CaM/Ca²⁺ stimulation of one subunit in the holoenzyme leading to phosphorylation of the Thr286 residue on a neighboring subunit. Thus, while T286 phosphorylation is a measure of CaMKII kinase activity, it is a special case because the substrate is captive and has unusual phosphorylation kinetics. Presumably, the purified CaMKII δ fragment used for these experiments was a dimer, but the same "intramolecular" phosphorylation scenario applies, and, for this reason, it might be better to measure phosphorylation of an added CaMKII substrate to determine ribociclib's IC₅₀. as was done for measurement of the potency CycD1/CDK4 inhibition (Figure S2e). The IC₅₀ for CycD1/CDK4 was 2 nM. i.e., 75 times more potent than towards CaMKII.

There seems to be some confusion here that we can be clearer about in our language:

1) The measurement of the IC₅₀ for ribociclib against CaMKII δ in figure 2b was performed with purified CaMKII δ and a synthetic peptide substrate.

2) The blot in Figure 2c is measuring T286 phosphorylation of endogenous CaMKII in cardiomyocytes, not purified protein.

We have attempted to clarify this in the revised manuscript.

4. Figure 2C: In the cardiomyocyte experiments, did the authors show that ribociclib reduced phosphorylation of known CaMKII sites in its target proteins, as well that of pT286?

We have tested phospholamban phosphorylation by CaMKII, which is reduced in the presence of ribociclib. We have added this as **Supplementary Fig. 3d** (see **Figure R3** below).

Figure R3. Ribociclib inhibits endogenous CaMKII activity. Protein immunoblot of human cardiomyocyte lysates that have been stimulated with Ca²⁺ in the presence of varying concentrations of ribociclib. Total CDK6 is shown as a control; pThr¹⁶/pSer¹⁷-phospholamban is shown as an indication of endogenous CaMKII activity.

5. With regard to CaMKII as a target for ribociclib, the Novartis group's in depth comparison of the specificities of ribociclib, palbociclib, and abemaciclib (Kim et al. Oncotarget 9:35226-35240, 2018) showed that of the ~400 kinases tested in a specificity screen CaMKII family members were the second most susceptible kinases to ribociclib inhibition after CDK4/6 - but at 100 nM ribociclib only 30% CaMKII inhibition was observed.

We thank the reviewer for pointing out this reference. 30% inhibition at 100 nM seems sensible, given our IC₅₀ measurement of ~150 nM. We have added this reference to the manuscript.

6. Figure 3: The authors could expand their comments on some of the observed interactors in panels B-D. For instance, the fact that the HopBF1 bait interacted most efficiently with the LTN1/listerin E3 Ub ligase, but it is not clear if LTN1 is this a known interactor. The p38 β MAP kinase interacted most strongly with the NCK1 adaptor protein, which they indicate is a known p38 β interactor. However, with regard to NCK1 being a p38 β substrate, Kinase Library predictions rank p38 β as the 29th most likely kinase to phosphorylate S91, the 47th most likely kinase to phosphorylate S166 and the 48th most likely to phosphorylate S262, the three reported Ser/Thr.Pro sites in NCK1, and p38 β is not listed among the 216 NCK1 interactors in BioGRID. Also, CSK, another known p38 β interactor does not seem to have any known Ser/Thr.Pro phosphosites, and as a dedicated SFK kinase it seems unlikely that CSK would phosphorylate p38 β . In addition, p38 β is not listed among the 871 CSK interactors in BioGRID.

1) For the putative HopBF1:LTN1 interaction, the average enrichment value is inflated largely due to one of the replicates being an outlier. We do not expect this to be a biologically relevant interaction, but we cannot rule it out and have no other justification for omitting this result.

2) This is a fair point; the p38 β interactome is quite complicated and overlaps with other kinases. We do not believe NCK1 to be a substrate of p38 β , and we agree that there is no evidence in the literature for CSK to be a direct interactor of p38 β . We have updated the figure and text to reflect this and to further specify known interactors and known substrates.

7. Figure 6: With regard to additional applications of SidBait, have the authors tested whether, with modifications to allow RNA tagging, SidBait could also be used to identify sequence-specific RNA-binding proteins?

We have not tested this technique with non-protein/small molecule baits, but we are excited about the prospect of expanding SidBait for other applications.

8. Page 8/9: Although the authors argue that SidBait's use in cell/tissue lysates is a

virtue, this could result in detection of spurious interactions due to loss of compartment boundaries, for instance upon nuclear and organelle lysis.

This is true, and a big reason for why we followed up on the ribociclib/RavB results to the extent that we did. We wanted to demonstrate that a hit from a target ID experiment is ultimately just a hypothesis and needs to be verified.

9. From a SidBait design perspective, would the inclusion and use of (Tev) protease cleavage sites in the Ub-bait linker beyond the Avi-tag biotinylation site and close to the bait protein give cleaner interactome results?

We have attempted various methods to dissociate the components of the bait from the targets, but we have found that the results are cleanest in the experiments with the fewest sample processing steps post-capture.

Dear Vincent,

We have now received re-review reports from two referees, which I have included below. As you will see, both referees maintain that there are further experiments that could be done. In particular, Referee #1 insists that a side-by-side comparison with a currently standard proximity labelling technique would have been both within scope and accessible. There are also concerns raised as to an undefined specificity of the ligase reaction. Despite these concerns, however, both referees state that the technical foundation of the manuscript is secure, and describe SidBait as a potentially useful addition to many labs' experimental repertoire. As such, we have decided that the manuscript should proceed towards publication. As part of this process, there are some remaining editorial points which need to be addressed. In this regard would you please:

- remove the figures from the main manuscript file,
- include up to five keywords,
- rename the "Materials and correspondence" section to "Data Availability",
- rename the conflict of interest statement to the "Disclosure and competing interests statement",
- remove the AC/CrediT section from the text,
- ensure that all figure callouts in the text are referred to sequentially,
- upload all figures as individual Figure files, with legends placed below the References; source file name, title, legend and manuscript callout all need to be updated to Figures EV1-EV7 instead of Supplementary Figure 1-7 (the term "Supplementary" should not be used for additional figures/data),
- update source file name, title, legend and manuscript callout to Dataset EV1-EV9 instead of Supplementary Table 1-9; legends should be correctly uploaded as separate tab/sheet in each Excel file, just need to be renamed,
- save the Appendix file in PDF format,
- include a Reagents and Tools table,
- upload high-resolution figures as well as high-resolution blot source data (Figure 5H, Appendix Fig. S1B and Appendix Fig. S4C are particularly affected),
- provide specific URLs for datasets 9BLH, 9BLI in the data availability statement,
- define 'n' in the legend of figure 4C,
- define the error bars in the legends of figures 2D and 4C,
- rename "Main Text" as "Introduction",
- remove the "This PDF file includes:" section from the manuscript, and
- correct the order and rename sections as follows: Title page - Abstract - Keywords - Introduction - Results - Discussion - Methods - Data Availability - Acknowledgements - Disclosure and Competing Interests Statement - References - Figure Legends - Table(s) - Expanded View Figure Legends.

We include a synopsis of the paper (see <http://emboj.embopress.org/>). Please provide me with a general summary image, a two sentence statement and 3-5 bullet points that capture the key findings of the paper.

I am looking forward to receiving your revised manuscript.

EMBO Press is an editorially independent publishing platform for the development of EMBO scientific publications.

Best wishes,

William

William Teale, PhD
Editor
The EMBO Journal
w.teale@embojournal.org

We realize that it is difficult to revise to a specific deadline. In the interest of protecting the conceptual advance provided by the work, we recommend a revision within 3 months (5th Feb 2026). Please discuss the revision progress ahead of this time with the editor if you require more time to complete the revisions. Use the link below to submit your revision:

Referee #1:

In general I felt that the authors did not sufficiently address my concerns.

Regarding my major point of comparing the method with the existing proximity based methods, authors did not address this. I do not agree that simple experiment is outside the scope of this study.

Another point of SidE PDE domain selectivity, Kalayil et al Figure 3C and Supplementary table 3 show that there are clearly some disallowed residues around target serine. And this analysis was even incomplete. There are likely more disallowed residues around Serine which simply was not tested comprehensively in Kalayil et al., This really touches on the heart of the method and It would be important to address this. To say that only serine is required for modification is wrong.

Finally regarding the results with Ribociclib experiments, it was important to do the binding affinity measurement experiments with the two kinases they found to understand the method's priorities in labeling. But unfortunately even this was not done.

While the method has a genuine potential, it is an incomplete story that contains a lots of bits and pieces of experiments that worked. While I feel no particular aspect has been thoroughly explored.

Referee #2:

In response to the reviewers' comments, the authors have done some additional experiments, included a new supplementary figure and some new panels and made a few textual changes to satisfy the reviewers' points. In general, these changes address this reviewer's concerns. While there is clearly more that can be done to validate and extend the utility of this system, SidBait is a clever new protein proximity ligation technology with some limitations but some advantages that seems likely to be adopted by the field, particularly for defining off-target interactions of small molecules.

All editorial and formatting issues were resolved by the authors.

Dear Vinnie,

I am pleased to inform you that your manuscript has been accepted for publication in the EMBO Journal.

Congratulations!

Best wishes,

Will

William Teale, PhD
Editor
The EMBO Journal
w.teale@embojournal.org

Please note that it is The EMBO Journal policy for the transcript of the editorial process (containing referee reports and your response letters) to be published as an online supplement to each paper. If you should prefer removal of any referee-only figures included in the point-by-point response(s), e.g. because they may still be used for future publication or because they have been reproduced from published work by others, please do let us know immediately via response email.

More information is available here: https://www.embopress.org/transparent-process#Review_Process
